# An antiplasmid system drives antibiotic resistance gene integration in carbapenemase-producing *Escherichia coli* lineages

Pengdbamba Dieudonné Zongo[1,2,3], Nicolas Cabanel[1,3,9], Guilhem Royer [1,3,9], Florence Depardieu [3,4], Alain Hartmann [5], Thierry Naas [6,7,8], Philippe Glaser [1,3,10] & Isabelle Rosinski-Chupin [1,3,10] ✉

Plasmids carrying antibiotic resistance genes (ARG) are the main mechanism of resistance dissemination in Enterobacterales. However, the fitness-resistance trade-off may result in their elimination. Chromosomal integration of ARGs preserves resistance advantage while relieving the selective pressure for keeping costly plasmids. In some bacterial lineages, such as carbapenemase producing sequence type ST38 *Escherichia coli*, most ARGs are chromosomally integrated. Here we reproduce by experimental evolution the mobilisation of the carbapenemase *bla*$_{OXA-48}$ gene from the pOXA-48 plasmid into the chromosome. We demonstrate that this integration depends on a plasmid-induced fitness cost, a mobile genetic structure embedding the ARG and a novel anti-plasmid system ApsAB actively involved in pOXA-48 destabilization. We show that ApsAB targets high and low-copy number plasmids. ApsAB combines a nuclease/helicase protein and a novel type of Argonaute-like protein. It belongs to a family of defense systems broadly distributed among bacteria, which might have a strong ecological impact on plasmid diffusion.

Antibiotic resistance is of major public health concern, having been responsible for more than 1.2 million deaths in 2019[1]. Mobile genetic elements, particularly plasmids carrying antibiotic resistance genes (ARG), are key contributors to antibiotic resistance spread within and between species[2]. While often carrying features beneficial under specific adverse environments, plasmids frequently confer a fitness cost to their bacterial host[3,4]. This is expected to lead to their loss over time and, under positive selection, to the integration of beneficial traits into the host chromosome as predicted by Bergstrom et al.' mathematical model of plasmid persistence[5]. However, some plasmids may persist over a long period even in the absence of selective pressure, a phenomenon known as the plasmid paradox[6,7]. Solutions to this paradox include compensatory mutations improving the fitness of plasmid-carrying bacteria[8–10] and high conjugation rates[11]. On the other hand, bacteria have developed defense mechanisms to counteract invasion by parasitic DNAs, some of them able to limit plasmid diffusion[12–14].

Carbapenemases are enzymes responsible for the hydrolysis of carbapenems, broad-spectrum antibiotics of major clinical importance.

[1]Ecology and Evolution of Antibiotic Resistance Unit, Institut Pasteur, Paris, France. [2]Sorbonne Université, Paris, France. [3]Université Paris Cité, Paris, France. [4]Synthetic Biology Unit, Institut Pasteur, Paris, France. [5]UMR AgroEcologie 1347, INRAe, Université Bourgogne Franche-Comté, Dijon, France. [6]Team ReSIST, INSERM UMR 1184, Paris-Saclay University, Le Kremlin-Bicêtre, France. [7]Department of Bacteriology-Hygiene, Bicêtre Hospital, APHP, Le Kremlin-Bicêtre, France. [8]Associated French National Reference Center for Antibiotic Resistance, Bicêtre Hospital, Le Kremlin-Bicêtre, France. [9]These authors contributed equally: Nicolas Cabanel, Guilhem Royer. [10]These authors jointly supervised this work: Philippe Glaser, Isabelle Rosinski-Chupin. ✉e-mail: isabelle.rosinski-chupin@pasteur.fr

In Western Europe, the class D enzyme OXA-48 is the most frequent carbapenemase in Enterobacterales[15–17]. This is generally attributed to the high conjugation rate of the IncL group pOXA-48 plasmids[18]. These 56–76 kb-long plasmids induce variable fitness costs to their hosts[19]. In addition, OXA-48 is also encoded on the chromosome[15,20,21], mainly in *E. coli* isolates of sequence type ST38, a major cause of extra-intestinal infections[22]. These ST38 isolates belong to disseminated clones with the $bla_{OXA-48}$ gene embedded in a complete or partial composite transposon, Tn*6237*, that likely derives from a pOXA-48 plasmid.

The factors driving ARG integration into the chromosome are largely unknown. Here, to identify such factors and to decipher the biological mechanisms leading to the selection of ARG integration, we experimentally characterize the move of the carbapenemase $bla_{OXA-48}$ gene from a pOXA-48 plasmid into the chromosome. We followed the integration of $bla_{OXA-48}$ through experimental evolution and show that the emergence of ST38 lineages with integrated $bla_{OXA-48}$ depends on the fitness cost imposed by pOXA-48 and on its elimination by a novel antiplasmid system.

## Results

### pOXA-48 plasmids induce a fitness cost and are unstable in ST38 *E. coli*

To set up an experimental model to study $bla_{OXA-48}$ integration we used *E. coli* ST38 because of its evolutionary history characterized by pervasive ARG integrations[23–26] in addition to its clinical relevance. We chose three ST38 isolates from water or sewage that did not carry $bla_{OXA-48}$ (Supplementary Data 1) and transferred into these isolates three variants of the IncL pOXA-48 plasmid (pOXA-48_1, pOXA-48_2 and pOXA-48_3) by conjugation. The three isolates differed by their position in ST38 phylogenetic tree (Supplementary Information Supplementary Fig. 1). In particular, ST38-1 belonged to a branch of the tree also encompassing three sub-clades characterized by different $bla_{OXA-48}$ chromosomal integrations[15] (Supplementary Information Supplementary Fig. 1). It is closely related to one of these sub-clades, colored in blue on the tree, sharing the same H-and 0-antigens (O2-H30). Plasmids were scarce in the three selected isolates. ST38_1 and ST38_3 contained only an IncFII and an IncFIC(FII) plasmid, respectively, whereas ST38_2 had no plasmid. Their complete genome sequence showed that, in ST38_1 and ST38_2, all ARGs were chromosomally inserted. In contrast, in ST38_3 most ARGs were carried by the IncFIC(FII) plasmid (Supplementary Data 1). We used as donor strains of the three pOXA-48 plasmids three clinical *Klebsiella pneumoniae* isolates. *K. pneumoniae* is the most common pOXA-48 plasmid-bearing species found in hospitals[27]. The three plasmids mainly differed by the presence and position of IS*1* generating or not a composite transposon, Tn*6237* (Fig. 1a). This 21.5 kb transposon consists of two IS*1* bracketing $bla_{OXA-48}$ and other plasmid genes. While this structure was present in pOXA-48_1, it was inactivated in pOXA48-2 by a 6.8 kb-long deletion, including the $bla_{OXA-48}$-distant IS*1*. In pOXA-48_3, the 4.5 kb region including $bla_{OXA-48}$ and the $bla_{OXA-48}$-proximal IS*1* was inverted, leaving $bla_{OXA-48}$ outside the composite transposon (Fig. 1a). We first showed that the plasmid structure did not influence the meropenem minimum inhibitory concentration (MIC) of the transconjugant unlike the genetic background of the recipient strain with a higher MIC for ST38-1 transconjugants, 0.5 μg ml-1, compared to 0.25 μg ml-1 and 0.38 μg ml-1 for ST38_2 and ST38_3 transconjugants respectively (Supplementary Information Supplementary Table 1).

To assess the fitness cost induced by pOXA-48 plasmids we compared the maximum growth rate of the transconjugants and isogenic plasmid-free strains. We estimated the relative fitness as the ratio of doubling time of transconjugants over plasmid-free strain. At least 15% doubling time increase was induced in lysogeny broth (LB) medium by pOXA-48 plasmids in ST38_1 (Fig. 1b). In contrast, in this medium, the three plasmids induced a lower (<5%) fitness cost in ST38_2 and ST38_3 (Fig. 1b). Fitness cost was dependent on the growth

medium as in M9 glucose minimal medium it rose up to 12% for ST38_2 transconjugants, while decreasing to 7–11% in ST38_1 transconjugants.

As pOXA-48 plasmids often conferred a fitness cost to their host, we quantified their stability in the three ST38 strains, following ten-day serial passages of the transconjugants (ca. 80 generations) in LB in the absence of antibiotic. The three pOXA-48 plasmids were gradually lost and at day 10, more than 95% of the population in ST38_1 and 40–80% in ST38_2 and ST38_3 had lost pOXA-48, irrespective of the plasmid variant (Fig. 1c).

### Twenty-eight days of in vitro evolution led to frequent $bla_{OXA-48}$ chromosomal integration in ST38_1

To determine whether the fitness of pOXA-48 transconjugants could be improved by plasmid-host coevolution, we first performed 28-day experimental evolutions of ST38_1/pOXA-48_1 transconjugants (Table 1). We hypothesized that this strain/plasmid combination might lead to chromosomal integration of $bla_{OXA-48}$ and pOXA-48_1 loss. Indeed, ST38_1 was phylogenetically close to sub-clades with chromosomal $bla_{OXA-48}$ (Supplementary Information Supplementary Fig. 1), the Tn*6237* structure in pOXA-48_1 had the characteristics of a mobile composite transposon and pOXA-48_1 induced a high fitness cost in ST38_1. Given the instability of pOXA-48_1, the growth medium was supplemented with subinhibitory concentration (0.1 μg.ml-1) of meropenem every passage or every third passage (Table 1 and Supplementary Information Supplementary Table 2). The ST38_1 plasmid-free strain was evolved as control. Five independent lineages were derived for each condition. Experimental evolution led to a rapid growth rate improvement for all transconjugant lineages under both meropenem conditions (Fig. 2a). To characterize the populations, we first performed pool-sequencing after 7, 14, 21 and 28 days of evolution. We observed a progressive decrease in read coverage of the pOXA-48_1 DNA-region outside Tn*6237* (Fig. 2b, Supplementary Information Supplementary Fig. 2), suggesting the enrichment of bacteria having lost pOXA-48_1 but keeping $bla_{OXA-48}$.

To further characterize putative transposition events of Tn*6237*, we PCR-screened meropenem-resistant colonies isolated at 28-day evolution for $bla_{OXA-48}$ and plasmid origin of replication (*repA*). We identified $bla_{OXA-48}^+$ *repA*- colonies in all ST38_1/pOXA-48_1 lineages (Supplementary Data 2). Two to three of these colonies per lineage were whole-genome-sequenced (WGS). Variant analysis revealed IS*1* new junctions in the chromosome or in the ST38_1 IncFII plasmid. We verified by PCR and Sanger sequencing that these new junctions corresponded to Tn*6237* transposition. In total, ten integrations at different positions in the chromosome and three in the IncFII plasmid were identified (Supplementary Data 3 and Fig. 2c). Analysis of pool sequencing reads revealed 44 additional new junctions, considering a threshold of 5% of the reads (Supplementary Data 4). Twenty were chosen for PCR analysis among which 16 were confirmed as Tn*6237* integration. Integrations were detected already at seven days of evolution and multiple integrations at multiple positions on the chromosome and the IncFII plasmid were selected in each lineage after 28 days (Supplementary Data 4 and Fig. 2c).

To determine whether $bla_{OXA-48}$ integration was associated with a fitness improvement, we selected three colonies, one in the chromosome and two in the IncFII plasmid. The growth rate of the three clones was increased with a 10–15% reduction in doubling time compared to the original transconjugant (Fig. 2d) and the meropenem MIC decreased by 50% (0.25 vs 0.5 μg ml-1) (Supplementary Information Supplementary Table 1). Therefore, in ST38_1, $bla_{OXA-48}$ integration appears as a major mechanism to relieve the fitness cost associated with pOXA-48_1 carriage.

### $bla_{OXA-48}$ integration depends on pOXA-48 structure and on recipient strain

To determine whether $bla_{OXA-48}$ integration during experimental evolution was dependent on the presence of Tn*6237*, we then evolved

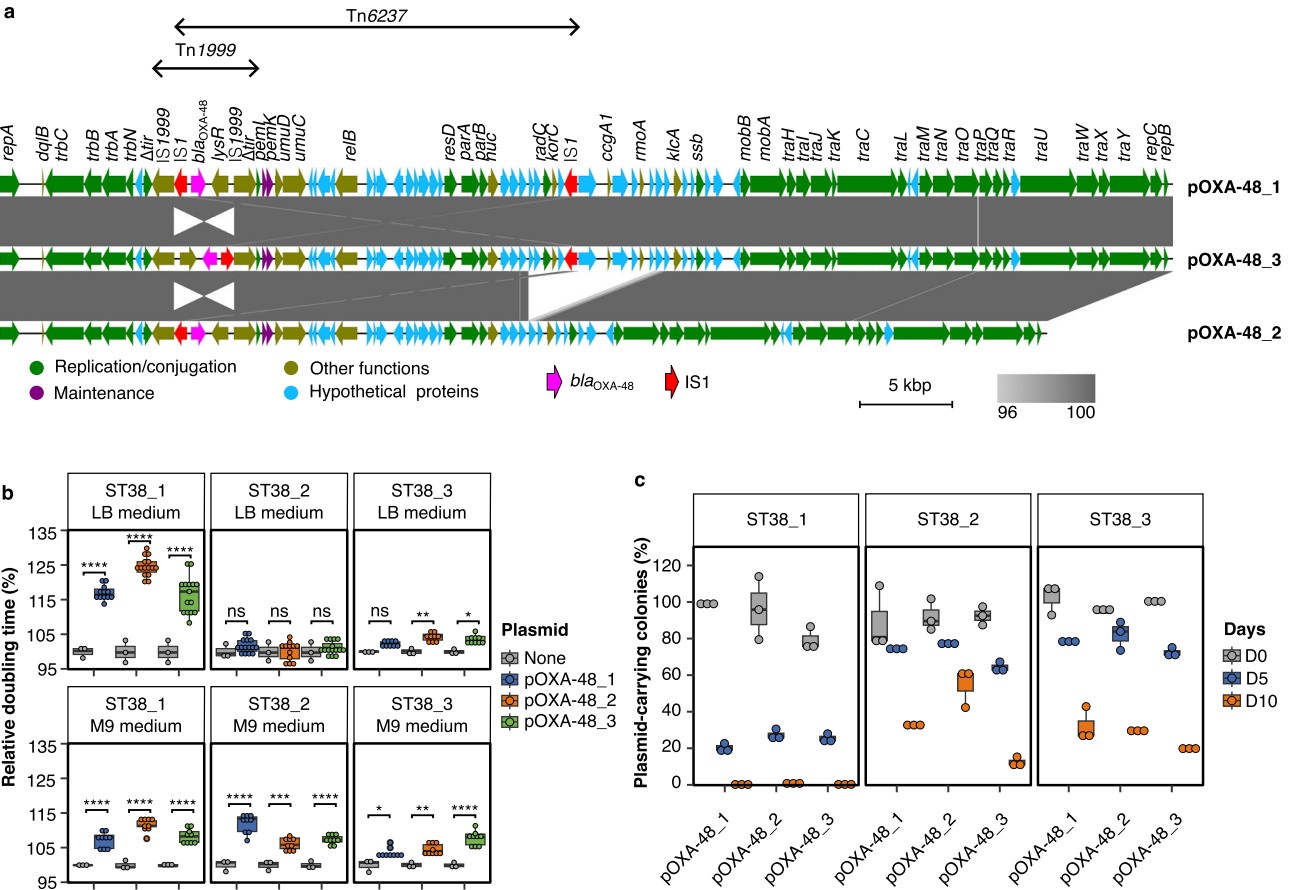

**Fig. 1 | pOXA-48 plasmids induce variable fitness costs and are unstable in three ST38 strains.** Three different variants of pOXA-48 were introduced from three *K. pneumoniae* isolates into three ST38 *E. coli* strains. **a** Alignment of the three pOXA-48 plasmids used in the study and displayed with Easyfig[59]. IS*1*s are indicated in red and the *bla*OXA-48 gene is in pink. The two composite transposons are indicated by double headed arrows. Tn*6237* carrying *bla*OXA-48 is only present in the pOXA-48_1 variant. Percentages of nucleotide identities are indicated by a gray gradient as indicated in the Figure key. **b** Comparison of the relative doubling time of plasmid-free strains and transconjugants calculated at exponential phase in the absence of meropenem in LB or M9 glucose minimum media. For each strain/plasmid combination, three to five transconjugants were tested. For each transconjugant, three independent experiments were performed. The relative doubling time is the ratio to the doubling time of the plasmid-free strain. **c** Plasmid loss during serial passages in the absence of meropenem. The ratio of plasmid-carrying colonies over the whole population at days 0, 5 and 10 was quantified by using the number of colonies growing on meropenem containing plates as a proxy for the number of

plasmid-carrying colonies. The results are from three biologically independent experiments. For boxplots in (**b**) and (**c**), the median is indicated by the line, the box bounds the 25th and 75th quartiles and whiskers bound the minimum and maximum values excluding outliers. Outliers are values > 1.5 × interquartile range. For (**b**), normal distribution of data was assessed with Shapiro-Wilk normality test. Statistical analysis was performed with pairwise two sample two-sided t-test and *p*-values were FDR-adjusted. Upper panel: Plasmid-free strains: *n* = 3; ST38_1 transconjugants: for pOXA-48_1: *n* = 12; for pOXA-48_2 and pOXA-48_3 *n* = 15; ****$p$ < 0.0001, $p$ = 1.58e-22, 3.71e-39, 2.35e-22 (from left to right); ST38_2 transconjugants *n* = 15 for each plasmid, ns: no significant difference; ST38_3 transconjugants: *n* = 9 for each plasmid; *$p$ < 0.05, **$p$ < 0.01, p = 9.97e-3, 2.9e-2 (from left to right). Lower panel: Plasmid-free strains: *n* = 3; for each strain/plasmid combination: n = 9; *$p$ < 0.05, **$p$ < 0.01, ***$p$ < 0.001, ****$p$ < 0.0001, $p$ = 1.02e-5, 1.05e-11, 4.72e-7, 1.85e-12, 2.34e-4, 9.27e-6, 2.9e-2, 8.67e-3, 4.12e-6 (from left to right). Source data are provided as a Source Data File.

ST38_1 transconjugants carrying plasmids pOXA-48_1, pOXA-48_2 or pOXA-48_3 in LB with meropenem added every third passage (Table 1). In contrast to pOXA-48_1, no fitness improvement throughout time was observed for pOXA-48_2, while the fitness of the population was slightly improved for pOXA-48_3 transconjugants at 28-day evolution (Fig. 3a). Bulk DNA sequencing of the whole populations at day 21 and at day 28 revealed after 28-day evolution only a relative decrease in read-coverage of the pOXA-48_3 region equivalent to the one lost in pOXA-48_1 (Supplementary Information Supplementary Fig. 3a, b) and a few new IS*1* junctions. Two Tn*6237* integrations in the chromosome were confirmed by PCR, suggesting the reconstitution of Tn*6237* in pOXA-48_3 (Supplementary Data 4). This showed that *bla*OXA-48 integration was dependent on its inclusion in Tn*6237*.

To determine whether the result of the experimental evolution was dependent on the ST38_1/pOXA-48_1 combination, we performed similar experimental evolution of ST38_2/pOXA-48_1 and ST38_3/pOXA-48_1 transconjugants (Table 1). For both experiments, we added

meropenem every third passage in LB. For ST38_2/pOXA-48_1 we also checked that a daily addition had no influence on the final result. In all cases, no evolution of the fitness was observed (Fig. 3b) while PCR-screening of meropenem-resistant colonies at 28-day did not identify any *bla*OXA-48+ *repA*- colonies (Supplementary Data 2). This suggested that in the absence of a sufficient fitness cost, *bla*OXA-48 integrations were not enriched enough to be detected. To test this hypothesis, we performed a similar experiment in minimal medium in which pOXA-48_1 induced a higher fitness cost to the ST38_2 transconjugants (Fig. 1b). A rapid improvement of growth rate was observed (Fig. 3c). We detected a few *bla*OXA-48+ *repA*- colonies, associated with a Tn*6237* chromosomal integration, in only two of the five evolved lineages by PCR screening (Supplementary Data 2 and Supplementary Data 3). These colonies showed a full fitness recovery (Supplementary Information Supplementary Fig. 3c). Pool sequencing did not reveal any integration site at a frequency superior to 5%, nor a significant plasmid loss (Supplementary Information Supplementary Fig. 3d). Therefore,

**Table 1 | Experimental scheme of the different evolution experiments**

| ST38 *E. coli* recipient strain | [a,b]Plasmid | Medium | MEM addition |
|---|---|---|---|
| **Experimental evolution 1 (Fig. 2):** Evolution with high fitness cost induced by pOXA-48 and blaOXA-48 in Tn*6237* | | | |
| ST38_1 | - | LB medium | No MEM |
| ST38_1 | pOXA-48_1 | LB medium | MEM / day |
| ST38_1 | pOXA-48_1 | LB medium | MEM / 3 days |
| **Experimental evolution 2 (Fig. 3):** Role of plasmid structure | | | |
| ST38_1 | - | LB medium | No MEM |
| ST38_1 | pOXA-48_1 | LB medium | MEM / 3 days |
| ST38_1 | pOXA-48_2 | LB medium | MEM / 3 days |
| ST38_1 | pOXA-48_3 | LB medium | MEM / 3 days |
| **Experimental evolution 3 (Fig. 3):** Evolution with low pOXA-48 induced fitness cost | | | |
| ST38_2 | - | LB medium | No MEM |
| ST38_2 | pOXA-48_1 | LB medium | MEM / day |
| ST38_2 | pOXA-48_1 | LB medium | MEM / 3 days |
| ST38_3 | - | LB medium | No MEM |
| ST38_3 | pOXA-48_1 | LB medium | MEM / 3 days |
| **Experimental evolution 4 (Fig. 3):** Evolution with intermediate pOXA-48 induced fitness cost | | | |
| ST38_2 | - | M9 medium | No MEM |
| ST38_2 | pOXA-48_1 | M9 medium | MEM / 3 days |
| **Experimental evolution 5 (Fig. 6):** Evolution in the absence of pOXA-48 destabilization by ApsAB | | | |
| ST38_1 | - | LB medium | No MEM |
| ST38_1Δ*apsAB* | pOXA-48_1 | LB medium | MEM / 3 days |

[a]Maps of the three pOXA-48 plasmids are shown in Fig. 1a.
[b]pOXA-48 plasmid were introduced by conjugation. A dash indicated the recipient strain with no pOXA-48 plasmid.

chromosomal integration and plasmid loss were not the main contributors to the fitness recovery observed in ST38_2 transconjugants, in contrast to ST38_1 transconjugants.

**pOXA-48 plasmids are stabilized by inactivation or mutation of a novel antiplasmid system**

In all evolved lineages, including those where Tn*6237* transposition was selected, bacteria still retaining the plasmid could be found after 28-day evolution. To identify other possible paths for plasmid-host coadaptation, we performed WGS of three to five *bla*$_{\text{OXA-48}}$$^+$ *repA*$^+$ colonies of each evolved lineage. We identified sporadic mutations occurring in the three ST38 evolved transconjugants and control plasmid-free strains, including only nine in different loci of pOXA-48 (eight in pOXA-48_1 and one in pOXA-48_2) (Supplementary Data 5). Mutations potentially decreasing susceptibility to carbapenems (in *ompC* or *envZ* for instance) were encountered. Large chromosomal deletions ranging from 15 to 40 kb and encompassing *mutS* and *rpoS* were also observed in evolved ST38_1 lineages. They were independent on pOXA-48 presence and meropenem addition, as they also occurred in control experiments. *mutS* loss led to a large number of mutations ($n = 13$ to $38$) likely resulting from an hypermutator phenotype. On the other hand, in 62% (86/138) of evolved transconjugants, we observed convergent evolution with mutations in two adjacent genes, F3141 or F3140, located in a ST38_1 genomic island (Fig. 4a) or its ortholog in ST38_2 (Supplementary Data 6). The most frequent mutations were IS*1* insertions ($n = 67$ with 37 different insertion sites). We also detected insertions of other IS ($n = 3$), non-synonymous ($n = 3$, two different), non-sense ($n = 2$), or frameshift ($n = 7$, four different) mutations and

complete or partial deletions ($n = 4$) of these genes (Supplementary Data 6). All (15/15) sequenced *bla*$_{\text{OXA-48}}$$^+$ *repA*$^+$ colonies from the M9-evolved ST38_2/pOXA-48_1 transconjugants carried mutations in orthologs of F3141 or F3140 and all ($n = 30$) from the five LB-evolved lineages shared the same IS*1* insertion in F3140 suggesting that it was present but not detectable in sequencing reads of the initial culture (Supplementary Data 6). No mutation in F3141-F3140 orthologs was detected in ST38_3/pOXA-48_1 transconjugants evolved in LB.

In ST38_1, six out of the ten tested mutations led to a fitness improvement compared to the original transconjugant, which was variable and always lower than following *bla*$_{\text{OXA-48}}$ integration and pOXA-48_1 loss (Fig. 4b). In ST38_2 the six tested mutations in F3141-F3140 orthologs led, in M9, to a full fitness recovery (Fig. 4b). Three ST38_1 and three ST38_2 evolved transconjugants mutated in F3140 or in F3141 were tested for pOXA-48_1 plasmid stability by 10-day serial passages in the absence of meropenem (Fig. 4c). In all tested transconjugants an increase in plasmid persistence was observed compared to the original transconjugant (Fig. 4c). pOXA-48_1 was kept in more than 90% of ST38-2 mutated transconjugants. In ST38-1 the stability was more variable with 50 to 100% bacteria keeping pOXA-48_1.

Conjugation of the pOXA-48 plasmids to genetically modified derivatives of ST38_1 and ST38_3 deleted for F3141-F3140 confirmed the stabilization of the plasmid in the absence of F3141-F3140 (Fig. 4d). Of note, ST38_2 was not amenable to genetic manipulation under the conditions we used. Furthermore, complementation in ST38_1ΔF3141-F3140 strain with a pHV7-derivative plasmid expressing F3141-F3140 under the control of the arabinose inducible promoter led to more than 80% loss of any of the three pOXA-48 plasmids at 48 h following induction (Fig. 4e). Induced expression of the operon with a non-sense mutation in either F3141 or in F3140 did not destabilize pOXA-48_1 confirming that the two proteins were needed for plasmid loss (Fig. 4f).

To determine whether F3141-F3140 might be involved in a more global antiplasmid activity, we compared the stability of five plasmids with different replication origins and copy numbers. In addition to the IncL pOXA-48 plasmids, F3141-F3140 deletion increased the stability of p15A, ColE1 and pMB1-type plasmids, but had no effect on pSC101 and IncFII/IncFIB plasmid stability (Fig. 5a). This confirmed that F3141-F3140 corresponds to a novel antiplasmid defense system that we renamed *apsAB* for antiplasmid system AB.

We found that ApsAB also reduces the conjugation frequency of the three pOXA-48 plasmids while comparing the transfer frequency with ST38_1 wild type or ST38_1 Δ*apsAB* as recipient strains (Fig. 5b). This suggests that ApsAB interferes with foreign DNA acquisition. However, ApsAB did not affect the transformation frequency of p15A and ColE1 plasmids (Fig. 5c). As some antiplasmid systems are also involved in antiphage defense, we tested the activity of ApsAB chromosomally expressed in MG1655 against eight different phages from *E. coli* (lambda, T4, P1, 186cIts, CLB_P2, LF82_P8, T5 and T7)[28] and did not observe any antiphage activity (Supplementary Information Supplementary Fig. 4).

Finally, to determine whether ApsAB actively eliminates plasmids, we quantified the persistence of a ColE1 plasmid over time following the induction of *apsAB* expression from a chromosomally integrated copy. Plasmid elimination was observed from between 2h30 and 3 h of arabinose induction and was almost complete ( > 95%) by a six-hour induction, whether or not chloramphenicol was added at 2h30 to stop cell division. These results indicate that ApsAB actively eliminates the plasmid, likely by promoting its degradation (Fig. 5d).

**ApsAB defense system is necessary for the selection of *bla*$_{\text{OXA-48}}$ integration**

We hypothesized that the elimination of pOXA-48 plasmids by ApsAB could contribute to the emergence of lineages with *bla*$_{\text{OXA-48}}$ inserted in the chromosome. To test this hypothesis, we deleted the *apsAB* operon in the original ST38_1/pOXA-48_1 transconjugant, which has

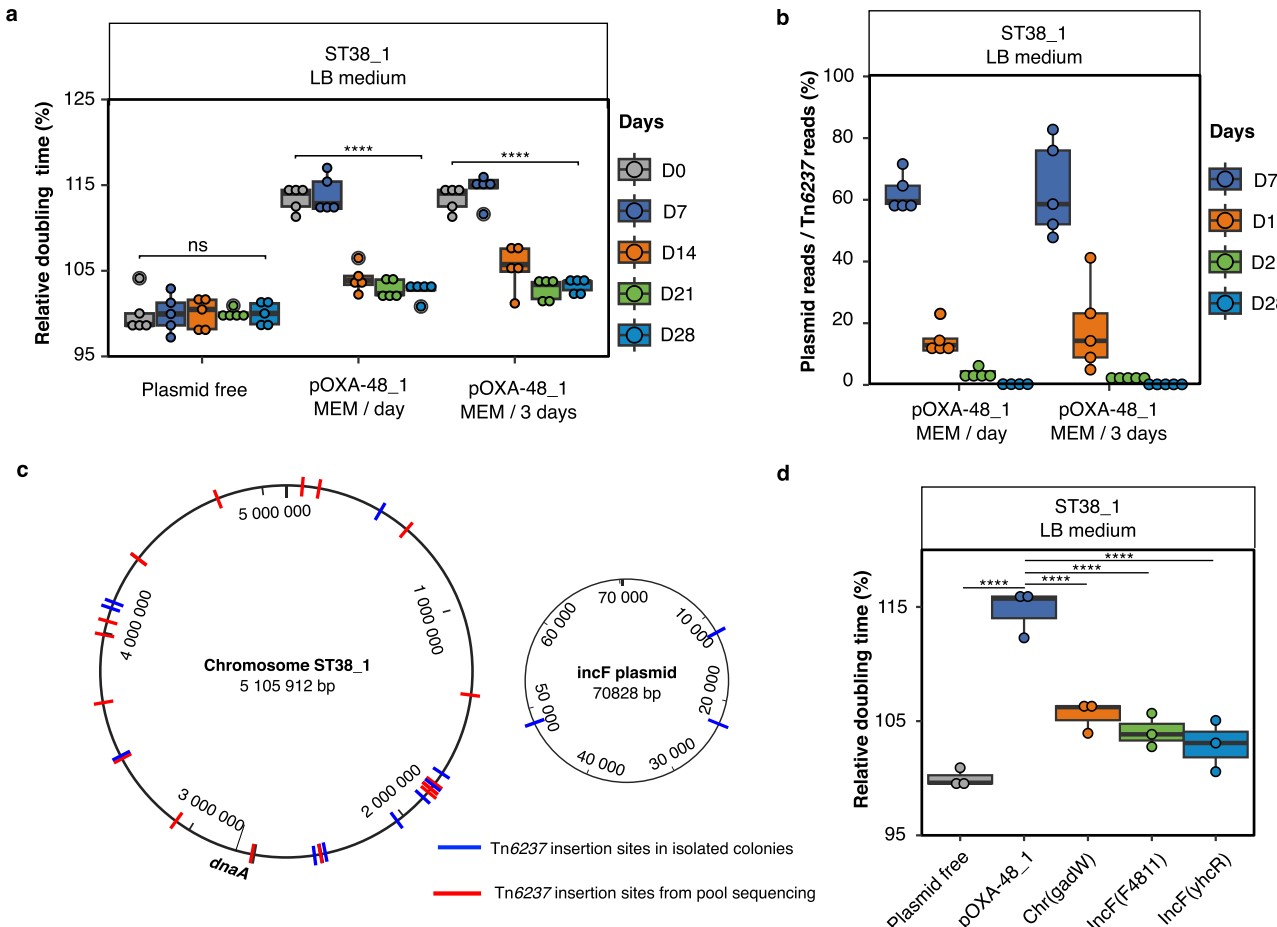

**Fig. 2 | Experimental evolution of pOXA-48_1 transconjugants selects Tn*6237* transposition and pOXA-48 loss in ST38_1.** Experimental evolution was performed by 28-day passages of ST38_1/pOXA-48_1 transconjugants or of the ST38_1 parental strain in LB. For transconjugants, meropenem was added every day (MEM/day) or every three days (MEM/3 days) in cultures at subinhibitory concentration. For each condition five lineages were evolved in parallel. **a** Evolution of the relative doubling time of ST38_1/pOXA-48_1 transconjugant population and of the parental strain as control. **b** Quantification of plasmid loss by WGS of whole populations for the evolved lineages in the two experimental conditions. Plasmid loss was estimated by calculating the ratio of the number of reads on three pOXA-48 regions of same size, two outside Tn*6237* and one inside (see also Supplementary Fig. 2). **c** Tn*6237*-insertion-sites identified in the evolved lineages. **d** Comparison of the relative doubling time of the original transconjugant and three $bla_{OXA-48}$ integrants (Tn*6237* integrated in the chromosome at *gadW* and at two positions in the IncFII endogenous plasmid. The integrant at *gadW* had an additional mutation in *malT*

frequently present in transconjugant and control evolved colonies. Chr: chromosome). In (**a**), (**b**) and (**d**), boxplots show median, box bounds 25th and 75th quartiles, whisker bounds minimum and maximum excluding outliers and outliers are values > 1.5 × interquartile range. For (**a**) and (**b**), the results are from five independent evolved lineages, for (**d**) they are from three biologically independent experiments. Doubling times were determined during exponential growth phase in the absence of meropenem. The relative doubling time was calculated as the ratio of the doubling time to one of the evolved plasmid-free lineages collected on the same day in (**a**) or to the plasmid-free strain in (**d**). Normal distribution of data was assessed with Shapiro-Wilk normality test. Statistical analysis was performed with pairwise two sample two-sided t-test and *p*-values (Source Data) were FDR-adjusted. ns: no significant difference; ****$p < 0.0001$. a: $p = 9.28$e-22 and 3.5e-20; d: $p = 1.4$e-08, 3.5e-06, 6.4e-07 and 2e-07 (from left to right). Source data are provided as a Source Data File.

the capacity to rapidly evolve towards Tn*6237* integration. No significant difference in fitness was detected between the deleted strain and the original transconjugant (Fig. 6a). As expected, *apsAB* deletion led to pOXA-48_1 stabilization (Fig. 6b). We then performed experimental evolution in LB medium of five lineages of ST38_1Δ*apsAB*/pOXA-48_1 (Table 1). Contrary to the original ST38_1/pOXA-48_1 (Fig. 2a), no fitness recovery was detected after 28 days of evolution of ST38_1Δ*apsAB*/pOXA-48_1 (Fig. 6c). No $bla_{OXA-48}^+ repA^-$ colonies out of 120 tested colonies at day 28 was detected by PCR screening. Similarly, pool sequencing of the whole population at days 7, 14, 21, and 28 showed no loss of pOXA-48_1 plasmid by analyzing plasmid read-coverage and no Tn*6237* transposition based on the PCR testing of the few new IS*1* junctions ($n = 4$) (Fig. 6d). Our results, therefore, show that plasmid destabilization through the activity of ApsAB is a key factor in the emergence of lineages with chromosomally integrated $bla_{OXA-48}$.

## ApsAB is the first characterized member of a broad family of Argonaute-like systems

Sequence similarity search against the NCBI nr public database by BLASTP revealed positive matches of ApsA and ApsB only with proteins of unknown function. Nevertheless, using similarity search based on structure predictions, we predicted in ApsA a central helicase domain with conserved residues characteristic of superfamily 2 helicase[29] and a C-terminal domain containing a PD-(D/E)XK- superfamily nuclease motif[30] (Fig. 7a). Directed mutagenesis showed that substitutions predicted to impair ATP hydrolysis (E533A) and helicase activity (K221A) completely abolished pOXA-48 plasmids destabilization while a mutation in the predicted nuclease active site (K1435A) had a partial effect (Fig. 7b and Supplementary Information Supplementary Fig. 5). On the other hand, structural modeling of ApsB revealed a loose structural similarity with prokaryotic Argonaute proteins (pAgos) acting as nucleic acid-guided endonucleases. However, ApsB lacked

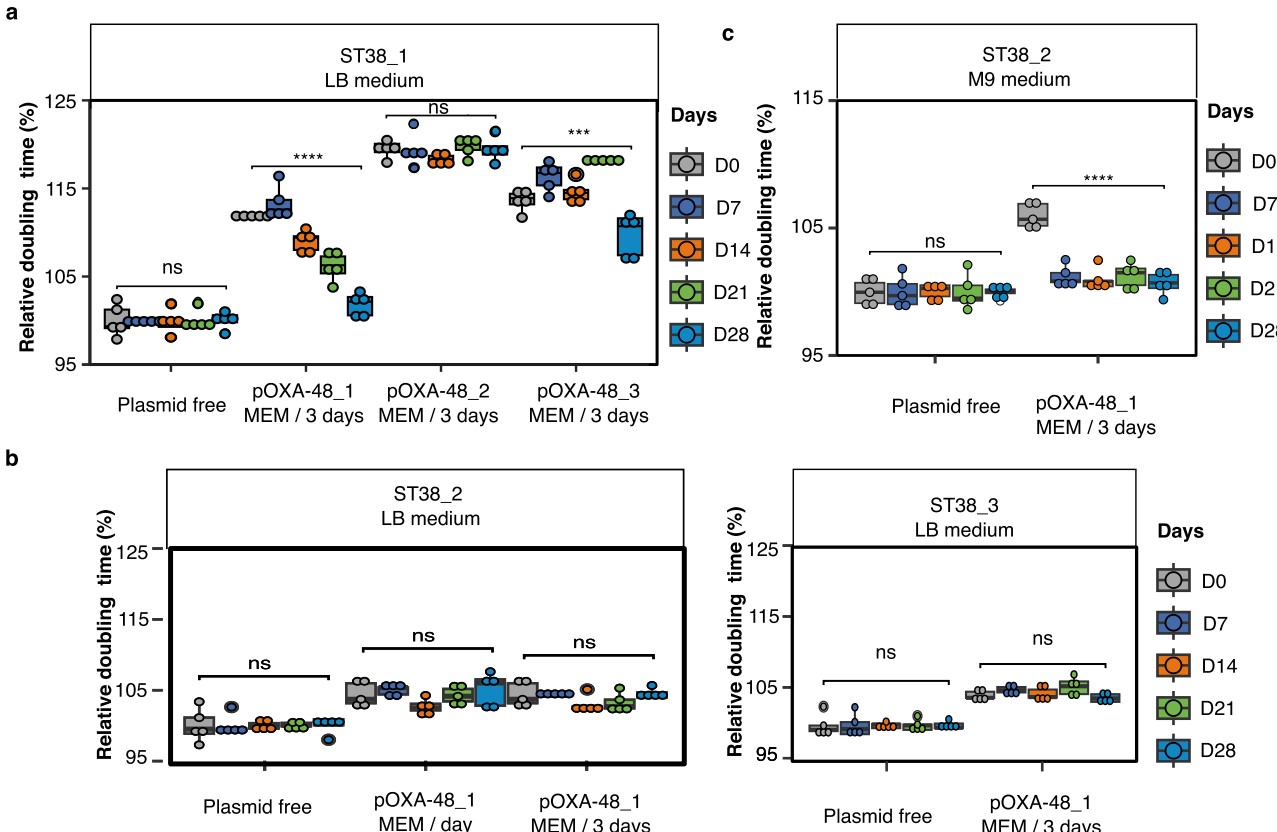

**Fig. 3 | Experimental evolution of pOXA-48_transconjugants depends on plasmid structure and plasmid-induced fitness cost.** Experimental evolution was performed by 28-day passages of transconjugants from the three ST38 *E. coli* strains in LB or in M9 minimal medium. For evolution of the transconjugants, meropenem was added every day (MEM/day) or every three days (MEM/3 days) in cultures at subinhibitory concentration. For each condition five lineages were evolved in parallel. **a** Evolution of the relative doubling time of ST38_1 pOXA-48_1, pOXA-48_2 or pOXA-48_3 transconjugant populations in the presence of meropenem at subinhibitory concentration and of the parental strain. **b** Evolution of the relative doubling time from ST38_2 and ST38_3 pOXA-48_1 transconjugants in the presence of meropenem at subinhibitory concentration. **c** Evolution of the relative doubling time of ST38_2 pOXA-48_1 transconjugants in M9 medium in the presence of meropenem at subinhibitory concentration. For (**a**), (**b**) and (**c**), doubling times were determined during exponential growth phase in the absence of meropenem. The relative doubling time was calculated as the ratio of the doubling time to one of the evolved plasmid-free lineages collected on the same day. Boxplots show median, box bounds 25th and 75th quartiles, whisker bounds minimum and maximum excluding outliers and outliers are values > 1.5 × interquartile range. For **a**, **b** and **c**, the results are from five independent evolved lineages. Normal distribution of data was assessed with Shapiro-Wilk normality test. Statistical analysis was performed with pairwise two sample two-sided t-test and *p*-values (Source Data) were FDR-adjusted. ns: no significant difference; ***$p < 0.001$, ****$p < 0.0001$. a: $p = 3.51e$-19, $p = 1.7e$-04 (from left to right); c: p = 8.17e-10. Source data are provided as a Source Data File.

typical PIWI and PAZ domains characteristic of Argonautes and was not identified among pAgos[31] (Fig. 7c). Sequence alignment of ApsB homologs retrieved from the NCBI database identified a conserved Y[X]$_3$K[X]$_n$QG[X]$_n$K motif (Fig. 7d, Supplementary Information Supplementary Fig. 6 and Supplementary Data 7), reminiscent of the conserved residues Y[X]$_3$K[X]$_n$Q[X]$_n$K characteristic of the MID domain of long-B pAgos that interacts with the 5'-end of guide nucleic acids[32]. The K413A substitution in this motif completely abolished pOXA-48 plasmid destabilization, supporting the requirement of this motif for ApsAB antiplasmid activity (Fig. 7b).

The association of an Argonaute-like protein and a protein with helicase and nuclease domains was reminiscent of the DdmDE plasmid defense system recently characterized in *Vibrio cholerae*[13]. No sequence similarity could be found between ApsB and DdmE. Their predicted 3D structures were different (Root-Mean-Square Deviation of atomic positions (RMSD) = 30.6) and DdmE did not contain the MID-like motif. However, searching for ApsA homologs by PSI-BLAST in databases unveiled a wide family of proteins ranging from ApsA-like to DdmD-like proteins. (Fig. 8a, Supplementary Data 8).

We identified ApsAB-like systems mainly among Enterobacterales but also in other gamma-proteobacteria and some beta-proteobacteria

and cyanobacteria (Fig. 8a and Supplementary Data 7, 8). In *E. coli*, complete or partially deleted *apsAB* homologs were located in at least three different families of genomic islands, some of them encoding other defense systems like Shango systems (Fig. 8b). We evaluated the antiplasmid activity of two representatives of *E. coli* ApsAB-like systems, showing 92/92 % or 28/25 % protein sequence identity with ST38_1 ApsA/ApsB respectively. Both systems were found to destabilize a ColE1 multicopy plasmid (Fig. 8c), indicating that the antiplasmid activity is not restricted to ApsAB from ST38 strains.

## Discussion

Mathematical modeling predicts that selective pressures, such as those exerted on bacteria during antibiotic treatment, will promote the integration of beneficial plasmid genes into the chromosome[5]. In this study, by investigating chromosomal integrations of *bla*$_{OXA-48}$, we demonstrate that, in addition to the selective advantage and a genetic structure promoting the mobility of the ARG, integration also depends upon a bacterial host antiplasmid system (ApsAB), here discovered. ApsAB influences plasmid-induced fitness cost and plasmid persistence. It belongs to a large family widespread across bacterial species ranging from cyanobacteria to Enterobacterales (Fig. 8a, Supplementary Data 7, 8).

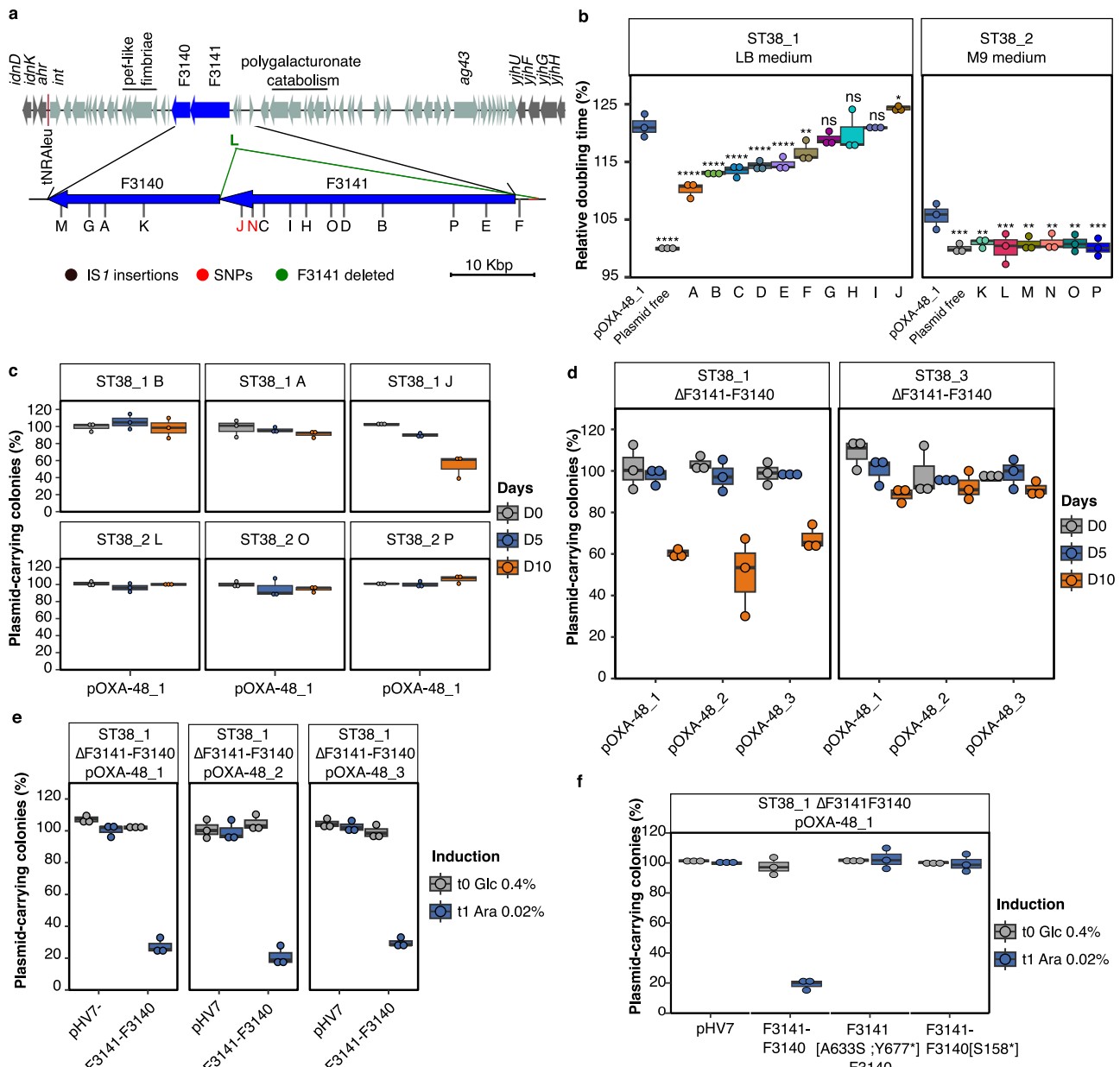

**Fig. 4 | Mutations in the F3141-F3140 operon led to pOXA-48 stabilization.**
**a** Organization of the genomic island containing the F3141-F3140 operon. The organization of the genomic island and its insertion at tRNA-leu is identical for the three ST38 strains. Positions of the mutations of the analyzed mutants named by letters are indicated by vertical dash. In black, IS*1* insertion, in green, a deletion of F3141 and in red, SNPs (strain J, L1444* and strain N, V1425W). **b** Doubling time comparison of ten F3141-F3140 mutant strains as indicated in (a) and of the original transconjugant. Doubling times were determined during exponential growth phase in the absence of meropenem in LB medium for ST38_1 strain and in M9 medium for ST38_2. The relative doubling time is calculated as the ratio to the doubling time of the plasmid-free strain. **c**, **d**, **e** and **f**. pOXA-48 stability in F3141-F3140 mutants after 10-day passages in LB medium in the absence of meropenem. **c** pOXA-48_1 stability in three ST38_1 and three ST38_2 mutants. Letters define the mutant strain as in (**a**) and (**b**). **d** Quantification of the carriage of the three pOXA-48 plasmids in ST38_1 and ST38_3 transconjugants deleted for the F3141-F3140 operon. and **f**. Quantification of pOXA-48 plasmids carriage following complementation of F3141-F3140

operon deletion. A plasmid copy of the complete F3141-F3140 operon (**e**) or of the operon inactivated for F3141 or F3140 (**f**) was expressed under the control of the p$_{BAD}$ inducible promotor in ST38_1ΔF3141-F3140 transconjugant. t0 refers to the value obtained following one 24-h passage in LB glucose (0.4%) and t1 to the value determined after two additional passages in LB arabinose (0.02%). pHV7-empty is used as negative control. For **a**, **b**, **c** and **d**, boxplots show median, box bounds 25$^{th}$ and 75$^{th}$ quartiles, whiskers bound minimum and maximum values excluding outliers and outliers are values > 1.5 × interquartile range. For **a**, **b**, **c** and **d**, the results are from three biologically independent experiments. For **b**, normal distribution of data was assessed with Shapiro-Wilk normality test. Statistical analysis was performed with pairwise two sample two-sided t-test and *p*-values (Source Data) were FDR-adjusted. ns no significant difference; **p* < 0.05, ***p* < 0.01, ****p* < 0.001, *****p* < 0.0001. b: LB medium: p = 3.12−18, 2.03e-09, 1.5e-6, 3.2e-6, 2.10e-5, 3.73e-5, 1.86e-3, 3.19e-2 (from left to right); M9 medium: *p* = 2.95e-4, 1.43e-03, 3.24e-4, 1.19e-3, 1.58e-3, 1.35e-3, 3.59e-4 (from left to right). Source data are provided as a Source Data File.

To investigate the factors involved in *bla*$_{OXA-48}$ gene integration into another replicon, we performed experimental evolutions of pOXA-48s transconjugants of three *E. coli* ST38 strains. Sub-inhibitory concentration of carbapenem was added every passage or

every three passages, leading to alternate selection for growth and for resistance (Table 1). Rapidly, for one of the five strain/plasmid pairs tested (ST38_1/pOXA48_1), *bla*$_{OXA-48}$ integrated lineages emerged and became dominant at day 28 (Fig. 2b). Tn*6237*

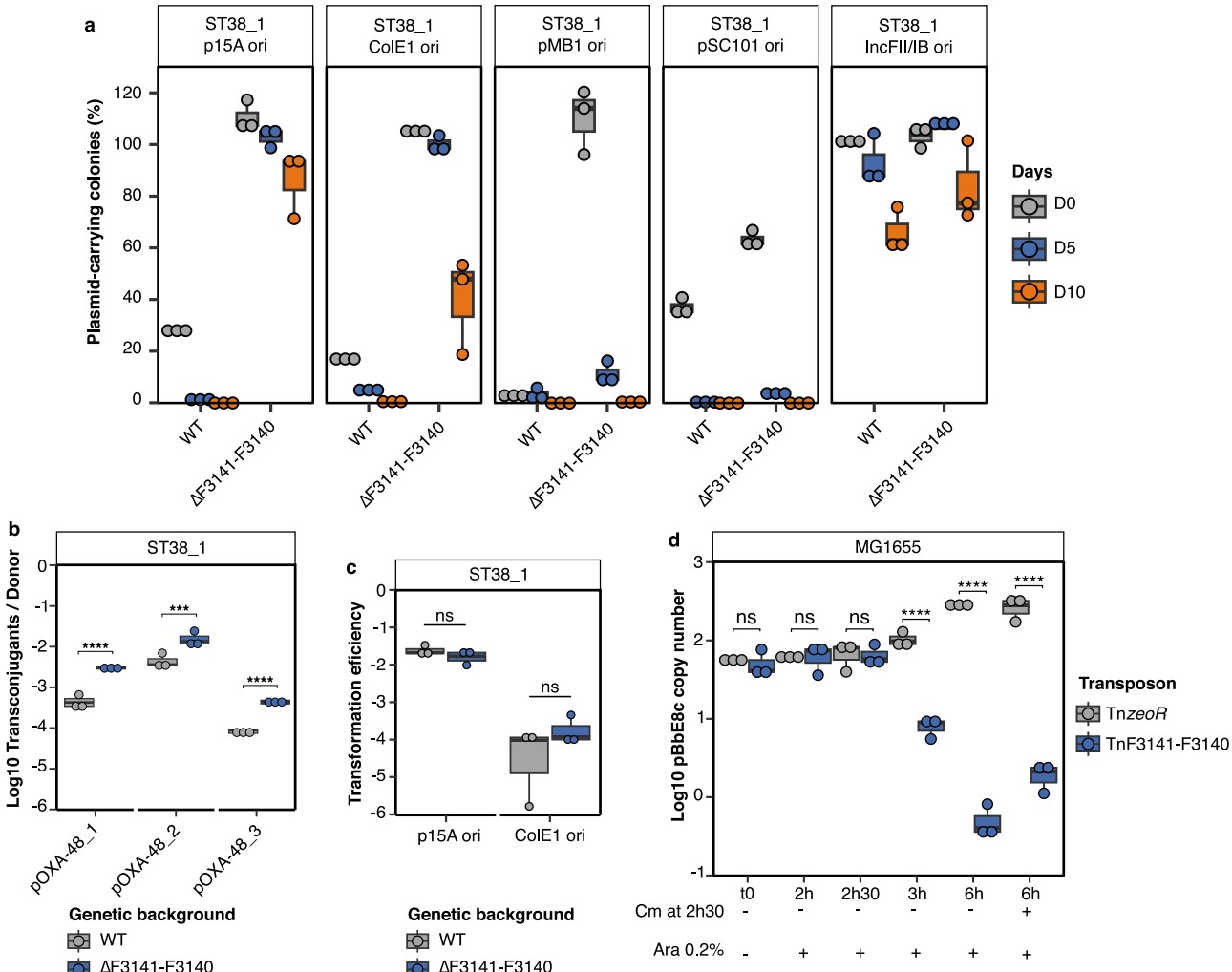

**Fig. 5 | F3141-F3140 (ApsAB) acts as an antiplasmid system active against low and high copy number plasmids. a** Activity of F3141-F3140 against non-IncL plasmids with different replicons and copy numbers: pACYt: p15A ori, ca. 10 copies; pBbE8c: ColE1 ori, ca. 20–30 copies; pUC19: pMB1 ori, >100 copies; pBbS8c: pSC101 ori, ca. 5 copies. CNR146C9-pKCP: incFII/IncFIB, ca. 1–2 copies **b** Comparison of pOXA-48 conjugation frequency between ST38_1 WT (wild type) and ST38_1ΔF3141-F3140. **c** Comparison of transformation efficiency into ST38_1 WT and ST38_1ΔF3141-F3140 by using electroporation of pACYt and pBbE8c **d** Quantification of the ColE1 derivative pBbE8c following induction of the F3141-F3140 operon cloned under the pBAD promoter and inserted into MG1655 *E. coli* chromosome (TnF3141-F3140). Tn*zeoR* used as control, corresponds to the empty transposon and only carries the zeocin resistance gene. The plasmid copy number was determined on DNA preparations by qPCR by using the ΔCt method with primers in pBbE8c replication origin and *rpsI* (chromosomal gene) as reference gene (Supplementary Data 9). For **a**, **b**, **c** and **d**, boxplots show median, box bounds 25th and 75th quartiles, whisker bounds minimum and maximum excluding outliers and outliers are values > 1.5 × interquartile range. For **a**, **b**, **c**, and **d**, the results are from three biologically independent experiments. For **b**, **c** and **d**, normal distribution of data was assessed with Shapiro-Wilk normality test. Statistical analysis was performed with Log10-transformed data by using a pairwise two-sample two-sided t-test and *p*-values (Source Data) were FDR-adjusted. ns no significant difference; ***$p < 0.001$, ****$p < 0.0001$. b: p = 0.0000198, 0.000648, 0.0000556 (from left to right). d: p = 4.15e-9, 4.96e-17, 8.47e-15 (from left to right). Source data are provided as a Source Data File.

transposition from pOXA-48_1 appears as particularly efficient to move $bla_{OXA-48}$ into the chromosome or the resident F-plasmid, as in each evolved lineage multiple integration events were selected. $bla_{OXA-48}$ chromosomal integration was also observed with pOXA-48_3, but at late time points. These integration events were also due to Tn*6237* transposition. We hypothesize that the Tn*6237* structure has been reconstituted following homologous recombination between the two IS*1999* copies surrounding $bla_{OXA-48}$ and the $bla_{OXA-48}$ proximal IS*1* (Fig. 1a). Altogether the experimental evolution successfully reproduced $bla_{OXA-48}$ integrations observed in worldwide disseminated ST38 clinical lineages[20,21]. Our results show that low doses of carbapenem, such as those encountered in the gut during parenteral administration of carbapenems[33], might have contributed to the emergence of $bla_{OXA-48}$ chromosomally integrated lineages.

In addition to $bla_{OXA-48}$ integration by transposition, we observed during experimental evolution convergent mutations (mainly inactivation) of the *apsAB* operon. ApsAB increases the fitness cost of pOXA-48 plasmids in ST38 transconjugants and destabilizes these plasmids (Fig. 4b, c). In experiments with ST38_1, the fitness gain linked to $bla_{OXA-48}$ integration and plasmid loss was high and lineages with $bla_{OXA-48}$ integration emerged rapidly. This was not the case with ST38_2 where both events led to equivalent fitness gain. This probably explains why *apsAB* inactivation events were generally selected at the expense of $bla_{OXA-48}$ chromosomal integration in this strain. More specifically, we showed that pOXA-48 plasmids destabilization was needed for the selection of $bla_{OXA-48}$ integration (Fig. 6d). *apsAB* location, embedded in a genomic island, and its distribution among ST38 isolates (Supplementary Information Supplementary Fig. 1) suggested that it has been horizontally acquired. This represents a typical

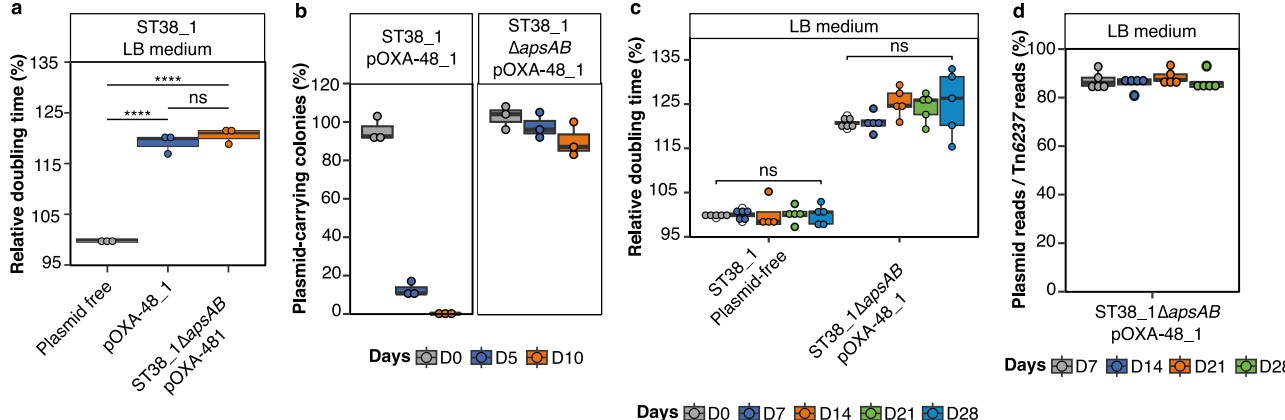

**Fig. 6 | $bla_{OXA-48}$ integration and pOXA-48_1 loss require the *apsAB* operon.**
**a** Comparison of the relative doubling time between the original transconjugant and ST38_1Δ*apsB* transconjugant **b** Quantification of plasmid carrying strains after *apsAB* deletion in the original transconjugant. **c** Evolution of the relative doubling time of ST38_1Δ*apsAB* transconjugant whole population and of the parental strain. **d** Quantification of plasmid loss by whole genome sequencing of whole populations (also see Supplementary Fig. 2). Plasmid loss was estimated by calculating the ratio of the number of reads on two pOXA-48 regions of same size, one outside Tn*6237* and one inside. Doubling times in (**a**) and (**c**) were determined during the exponential growth phase in the absence of meropenem. The relative doubling time is calculated as the ratio to the doubling time of one of the evolved plasmid-free lineages from the same day for (**c**) or to the plasmid-free strain in (**a**). For **a**, **b**, **c** and **d**, boxplots show median, box bounds $25^{th}$ and $75^{th}$ quartiles, whisker bounds minimum and maximum excluding outliers and outliers are values > 1.5 × interquartile range. The results are from three biologically independent experiments (**a** and **b**) and five independent evolved lineages (**c** and **d**). Normal distribution of data was assessed with Shapiro-Wilk normality test. Statistical analysis was performed with Log10-transformed data by using a pairwise two-sample two-sided t-test and *p*-values (Source data) were FDR-adjusted. ns no significant difference. ****$p < 0.0001$. a: p = 0.00000584, 0.00000584 (from left to right). Source data are provided as a Source Data File.

case of MGE negative interactions between a genomic island and plasmids[34,35]. *apsAB* was present in the three strains of the study and was associated to a low stability of the three pOXA-48 plasmids. However, while *apsAB* was frequently inactivated in colonies that retained pOXA-48 in ST38_1 and ST38_2, we did not detect inactivation events in ST38_3, suggesting that conservation of *apsAB* might be positively selected for an unknown reason in this genetic background. Nevertheless, *apsAB* deletion in this strain led to a stabilization of pOXA-48 plasmids (Fig. 4d). In addition to ApsAB, nine to eleven other defense systems were predicted in the three strains (Supplementary Data 1). None was specific to ST38_1 and we did not detect any mutation in these systems during experimental evolutions. Therefore, they probably did not influence the evolution results. *apsAB* is present in the four ST38 clinical lineages in which $bla_{OXA-48}$ is integrated in the chromosome (Supplementary Information Supplementary Fig. 1)[15] and has likely driven the integration event leading to their dissemination. However, as ARGs integrated in the genome are frequently being reported in resistant *E. coli* and *apsAB* homologs only found in a subset of strains, other plasmid destabilizing systems are probably to be involved. Alternatively, in the absence of an active antiplasmid system, the mobilization to the chromosome might be much less frequently selected.

In recent years, a plethora of bacterial defense systems have been discovered. But, only a few of them, such as CRISPR, restriction-modification, pAgos, DdmDE and Wadjet were described to have antiplasmid activity[13,36,37]. By combining different in silico search strategies, we uncovered a broad family of ApsA homologs that combine helicase and nuclease domains. Strikingly, this family was subdivided into two classes associated with two families of argonaute-like proteins, whose prototypes were ApsB and the recently described DdmE[13]. We therefore connected DdmDE- to ApsAB-like systems. However, unique to ApsB-like proteins is the conservation of a MID-like domain of long pAgos, which we showed to be essential for its activity. This suggests that ApsB is involved in a guide-dependent recognition of the target, ApsA bringing the nuclease activity, similarly to what has recently been described for a group of catalytically inactive long true pAgos[38]. While these Argonaute systems confer immunity also via

abortive infection[38], we do not have evidence that ApsAB kills plasmid invaded cells. Despite the absence of similarity between DdmE and ApsB, both DdmDE and ApsAB share an antiplasmid activity, like a second *E. coli* system, only 28/25% similar to ST38 ApsAB (Fig. 8c). This suggests that the compendium of systems we brought out represents a new diverse family of defense systems acting on plasmids. Their precise functional mechanisms, in particular the nature and specificity of the guides, remain to be characterized. These systems might also provide new tools for genomic engineering and plasmid curing, including those carrying ARGs.

To our knowledge, we provide the first evidence of an antiplasmid defense system influencing the evolutionary fate of an antibiotic resistance gene. Conversely, during our analysis of *apsAB* homologs, we retrieved many partially deleted systems, which might reflect the counter-selection of these systems in a context of antibiotic pressure that favors the stabilization and dissemination of ARG carrying plasmids.

## Methods

### Bacterial strains, growth conditions, and plasmids
Characteristics of the strains used in this work are summarized in Supplementary Data 1. All strains were submitted to Illumina sequencing and for some of them to long-read sequencing. Donor strains in conjugation experiments were three clinical *K. pneumoniae* isolates carrying either pOXA-48_1, pOXA-48_2 or pOXA-48_3. The organization of IS*1* sequences in the three plasmids was confirmed by PCR. Recipient strains were three *E. coli* ST38 environmental isolates characterized by different O-types/H-types. Their position in the ST38 phylogeny is given in (Supplementary Information Supplementary Fig. 1a). ST38_1 carried an endogenous IncFII, 70.8 kb-long plasmid deprived of any ARG while ST38_3 contained a 161 kb-long IncFIC(FII) plasmid. ST38_2 has no plasmid (Supplementary Data 1). *E. coli* strains CNR36C9 (ST219) and CNR81D10 (ST10) encoding closely related (91%/92% amino acid (a.a.) identity) or distantly related (28%/25% a.a. identity) *apsAB*-like systems respectively were used for PCR-cloning of *apsAB* homologs. *E. coli* K-12 strain MG1655 was used for *apsAB* expression experiments from a mini-Tn*7* inserted downstream *glmS*. *E.*

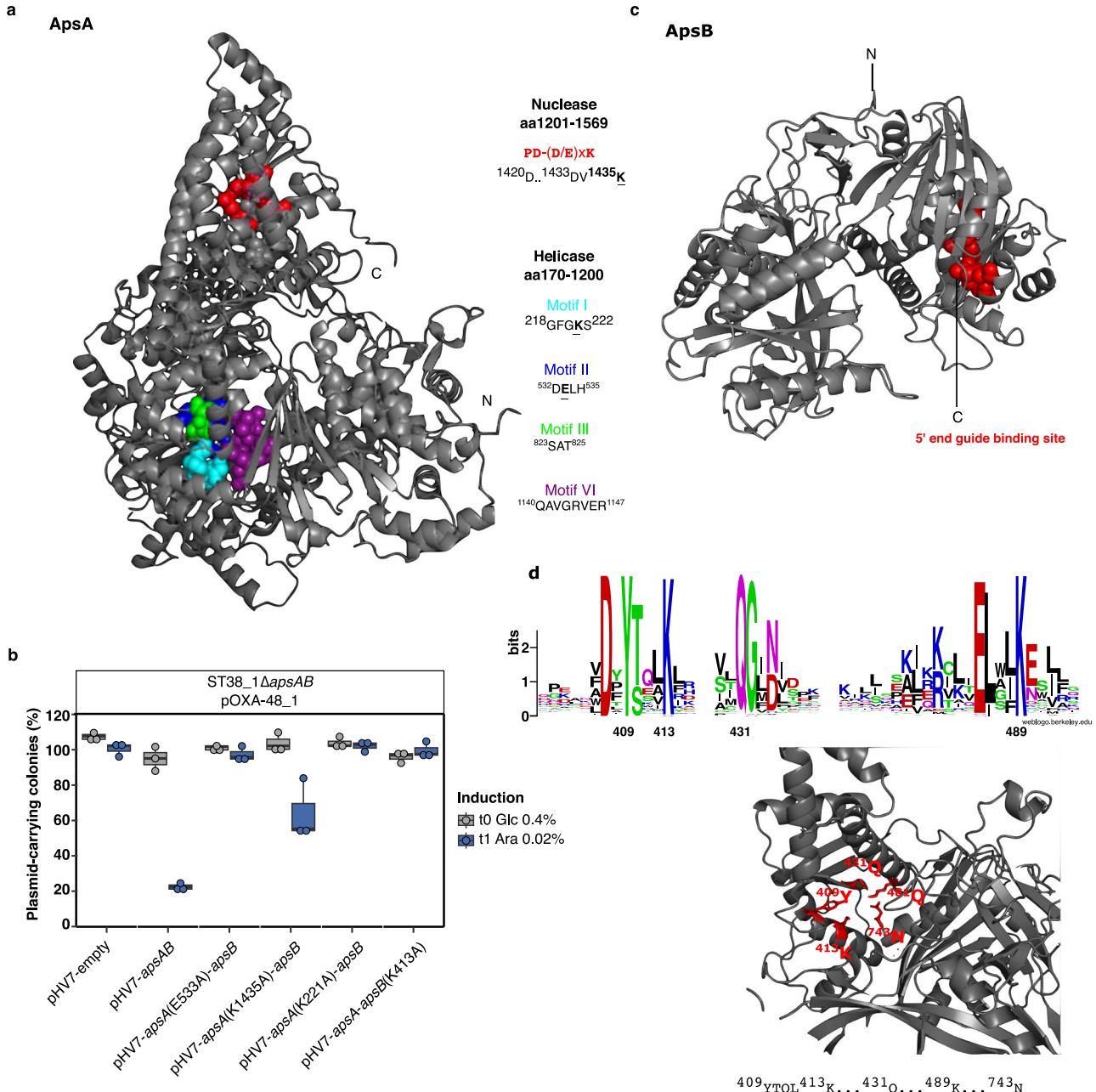

**Fig. 7 | ApsAB associates a protein with helicase and nuclease domains and a novel Argonaute-like protein. a** Predicted structure of ApsA as determined with Alphafold2. The conserved PD-(D/E)XK nuclease domain with active site residues ($^{1420}$D…$^{1433}$DV$^{1435}$K) in red and the four motifs of the helicase domain in cyan $^{218}$GFGKS$^{222}$, blue $^{532}$DELH$^{535}$, gray $^{823}$SAT$^{825}$ and purple $^{1140}$QAVGRVER$^{1147}$ are indicated. **b** Effect of mutations replacing key residues of the putative helicase (K221A, E533A) or nuclease (K1435A) domains of ApsA and of the putative 5′ end guide binding motif (K413A) of ApsB on pOXA-48_1 maintenance. Effect on pOXA-48_2 and _3 are shown in Supplementary Fig. 5. *apsAB* and variants were expressed under the control of the p$_{BAD}$ inducible promotor in ST38_1ΔF3141-F3140 transconjugants. t0 refers to the value obtained following one 24-h passage in LB glucose 0.4%, and t1 to the value determined after two additional passages in LB arabinose 0.02%. pHV7-empty (pHV7) is a negative control. Results are from three biologically independent

experiments. Boxplots show median, box bounds 25$^{th}$ and 75$^{th}$ quartiles, whisker bounds minimum and maximum excluding outliers, and outliers are values > 1.5 × interquartile range. Source data are provided as a Source Data File. **c** Predicted structure of ApsB as determined with Alphafold2. The conserved motif acting as 5′-end guide binding motif in the MID domain of pAgos, $^{409}$Y,$^{413}$K,$^{431}$Q, $^{489}$K, is indicated as red balls; the RNAse H fold of the PIWI domain and the PAZ domain of pAgos are not conserved in ApsB. **d** Sequence logo generated by WebLogo from the alignment of ApsB homologs (shown in Supplementary Fig. 6) in the region corresponding to the MID domain, highlighting the conserved residues. A zoom on the ApsB MID domain structure at the $^{409}$Y,$^{413}$K,$^{431}$Q, $^{489}$K residues is provided in the lower panel. The C-terminal $^{743}$N residue possibly interacting with the putative guide nucleic acid in the binding pocket is also indicated.

*coli* strains DH5a and XL1 blue were used for cloning and MFD *pir*[39] for propagation of plasmids with RK6 origin of replication and for bacterial mating.

Bacterial growths were performed at 37 °C. Liquid cultures were performed in LB Miller or in M9 medium supplemented with glucose

0.4%, MgSO$_4$ 1 mM and CaCl$_2$ 0.1 mM with shaking. Where appropriate, antibiotics were added: meropenem (0.1 μg ml-1), apramycin (40 μg ml-1), zeocin (30 μg ml-1) and chloramphenicol (10 μg ml-1). Growth and mating experiments with MFDpir derivatives were performed in LB or LB agar complemented with 0.3 mM diaminopimelic

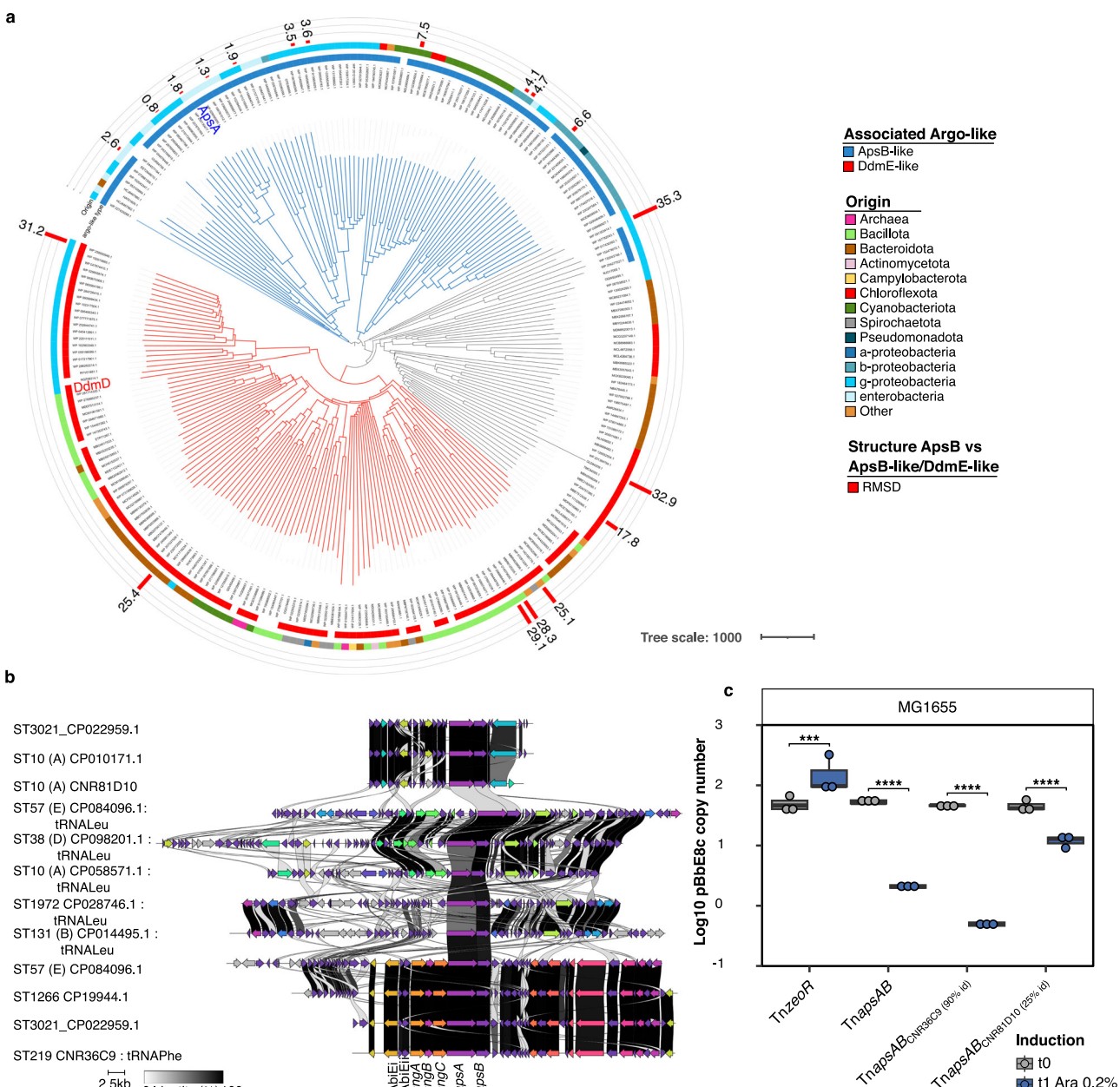

**Fig. 8 | ApsAB belongs to a broad family of putative antiplasmid systems.**
**a** Distance tree of ApsA related sequences retrieved from the NCBI database. Protein sequences were retrieved by using PSI-BLAST and the distance tree was obtained by the Neighbour-joining method. The distance tree separates proteins more related to ApsA (in blue) or to DdmD (in red), or intermediate (in gray). Circles are annotated as the figure key. From inside to outside: the type of associated Argonaute-like protein (ApsB or DdmE), the taxonomic origin of the sequence, and for 19 ApsB-like or DdmE-like proteins with a predicted 3-D structure in Uniprot or NCBI databases, a measure of their structural similarity with ApsB by using the root-mean-square deviation of atomic positions (RMSD)(See also Supplementary Data 8). **b** Alignment of different families of genomic islands containing a complete or a truncated *apsAB*-like operon. Other defense systems, such as the SngABC Shango system and the AbiEIAbiEII systems, encoded in some of these genomic islands are indicated. The sequence type (ST) and the strain ID are indicated, and the *E. coli* phylogroup is given between brackets. When it could be identified, the tRNA in which the genomic island is inserted is also indicated. *apsAB* variants from CNR36C9 and CNR81D10 tested in (**c**) have been included. **c** Quantification of the

ColE1 derivative pBbE8c following induction of two homologs of the *apsAB* operon encoding proteins with 92/92% and 28/25% aa sequence identity with ApsA/B, respectively. Operon was cloned in Tn*7* following PCR amplification from the ST219 CNR36C9 and ST10 CNR81D10 isolates under the p$_{BAD}$ promotor and inserted in MG1655 *E. coli* chromosome. Tn*ZeoR* used as control, corresponds to the empty transposon and only carries the zeocin resistance gene. The plasmid copy number was determined by qPCR by using the ΔCt method with primers in pBbE8c replication origin and *rpsl* (chromosomal gene) as the reference gene. The results are from three biologically independent experiments. Boxplots show median, box bounds 25th and 75th quartiles, whisker bounds minimum and maximum excluding outliers and outliers are values > 1.5 × interquartile range. The normal distribution of data was assessed with Shapiro-Wilk normality test. Statistical analysis was performed with Log10-transformed data by using a pairwise two-sample two-sided t-test and *p*-values (Source Data) were FDR-adjusted. ***$p < 0.001$, ****$p < 0.0001$.b: p = 4.69e-04, 1.32e-09, 1.77e-11, 8.71e-05 (from left to right). Source data are provided as a Source Data File.

acid (DAP, ThermoScientific). Induction from $p_{BAD}$ promoter was performed in LB Miller supplemented with L-arabinose to a final concentration of 0.02 or 0.2% as indicated. E-test were performed on Mueller Hinton agar (MHA) medium. A list of plasmids used in the study and their main characteristics is provided in (Supplementary Data 9). All inserted fragments were verified by Sanger sequencing (Eurofins Genomics), and expression plasmids carrying *apsAB* were WGS (PlasmidSaurus or Eurofins Genomics).

## Conjugation assay

Overnight precultures in LB of donors and recipient strains were diluted 1:100 into fresh LB and grown to an optical density at 600 nm ($OD_{600}$) of 0.6. After mixing at 1:1 ratio, 200 μl were spread on a filter (MILLIPORE type HAEP 0.45 μM) placed on LB agar and incubated at 37 °C overnight or for one hour. Bacteria were harvested in physiological water followed by serial dilutions and plated on LB agar containing two antibiotics: meropenem (MEM) 0.1 μg ml-1 and tetracycline (TET) 10 μg ml-1 to select transconjugants. Transconjugants were WGS and those devoid of mutations were selected for further analysis. Conjugation frequency was calculated as the ratio of transconjugants over donor after a 1h-mating followed by selection of transconjugants, donors, and recipients (MEM 0.1 μg ml-1 and TET 10 μg ml-1; MEM 0.1 μg ml-1, TET 10 μg ml-1 respectively).

## Growth curves and relative fitness assessments

Overnight precultures in LB or M9 of the three ST38 *E. coli* plasmid-free isolates and their isogenic transconjugants were inoculated without and with antibiotic (MEM 0.1 μg ml-1) respectively. 96-well plates were inoculated with 100 μL of precultures diluted to $5*10^5$ colony forming unit (CFU) per milliliter in LB or M9 medium and incubated at 37 °C with shaking for 6 (LB) or 16 h (M9) in an automatic plate reader (Tecan infinite M Nano under i-control 2.0.10). Three independent experiments were carried out for each strain. For each independent biological replicate, the doubling time was obtained by calculating the mean from three to five technical replicates. The doubling time was calculated using the formula: $G = \ln (2)/\mu_{max}$. $\mu_{max}$ is the maximum growth rate, corresponding to the slope of the curve at exponential phase. The relative doubling time was calculated using the formula: $W = G_{transconjugant}/G_{plasmid-free}$.

## Experimental evolution

Overnight precultures in LB or in M9 of transconjugants and plasmid-free strains were diluted 1:200 into 10 ml fresh LB medium or into complemented M9 medium with and without MEM 0.1 μg ml-1 respectively and incubated at 37 °C with shaking (220 r.p.m., INFORS HT Minitron). For each experiment, five independent biological replicate cultures were evolved. Serial transfers were achieved for 28 days using 1:200 dilutions every day into 10 ml of fresh medium (ca. 8 generations per day). For the evolution of transconjugant lineages, MEM was added every day or every three days at 0.1 μg ml-1 as indicated. Each seven days, whole populations were collected and frozen at −80 °C, and bacterial pellets for DNA sequencing were obtained by centrifuging 1 ml of culture. Relative fitness of whole population through experimental evolution was monitored by growth curves. PCR using primers (Sigma-Aldrich) targeting $bla_{OXA-48}$ and *repA* (Supplementary Data 9) were performed on isolated colonies selected on MEM 0.1 μg.ml-1 to identify potential integration of $bla_{OXA-48}$ (Tn*6237*) and pOXA-48 loss at day 7, day 14, day 21 and day 28.

## Plasmid stability assay

Overnight precultures of pOXA-48 transconjugants in LB with MEM 0.1 μg ml-1 were set as day 0. Plasmid stability was assessed by serial passages using 1:200 dilutions into 10 ml fresh LB without MEM for ten days. Bacteria were collected and frozen at −80 °C at days 0, 5 and 10 and CFU were determined by plating 100 μl diluted cultures on LB agar

with and without MEM 0.05 μg ml-1. Colonies growing on MEM were used as a proxy for plasmid-carrying bacteria as Tn*6237* integration was estimated as a rare event under these conditions. Automatic bacterial colony counting was done with scan4000 (INTERSCIENCE) and plasmid stability was calculated as the ratio of plasmid-carrying bacteria over total population. Due to dilution and plating biaises this might occasionally result in ratios slightly superior to 100%. The stability of other plasmids was similarly tested after introduction by electroporation (pBbS8c, pBbE8c, pACYt, pUC19) or conjugation (pKPC, using CNR146C9 as donor) with adequate antibiotic selection (Supplementary Data 9).

## Plasmid constructions for expression of wild-type and mutated copies of *apsAB*

F3141-F3140 operon, F3140 (*apsB*) and F3141 (*apsA*) were cloned under the control of a $p_{BAD}$ inducible promotor in a pHV7 vector coding for apramycin resistance. All growing steps of pHV7-F3141-F3140 transformants were performed in the presence of glucose 0.2 % to repress *apsAB* induction. Two mutated versions of this operon where a stop codon was introduced in F3141 or F3140 sequence were generated during this cloning and the corresponding plasmids were used for activity testing. For site-directed mutagenesis, selected codons were modified in the pHV7-F3141-F3140 plasmid by using the Q5 Site-Directed Mutagenesis kit (New England Biolabs) according to manufacturer's recommendations. pHV7 plasmid mutants were WGS (Eurofins). For chromosomal expression of *apsAB* (F3141-F3140) in MG1655, *apsAB* or *apsAB* homologs were cloned under the control of a $p_{BAD}$ promoter in a miniTn*7* transposon (miniTn*apsAB*) using primers described in Supplementary Data 9. miniTn*apsAB* was integrated downstream *glmS*, a neutral chromosomal position of *E. coli* K12 MG1655 strain following triparental mating as previously described[40]. Antiplasmid activity of these constructs was analyzed by electroporating the ColE1 plasmid pBbE8c. An MG1655 strain harboring an empty miniTn*7* (miniTn*zeoR*) integrated at *glmS* was used as control. When stated, chloramphenicol was added at 10 μg.ml-1 to arrest bacterial growth 2.5 h after addition of arabinose 0.2%.

## F3140-3141 (*apsAB*) chromosomal deletion and complementation

Complete deletion of the F3140-3141 (*apsAB*) operon was obtained by λ Red recombination[41] by using an in-lab p15red vector carrying the lambda recombinase under the control of an inducible pBAD promotor (Supplementary Data 9). Primers (Sigma-Aldrich) used for PCR amplification of the ZeoR cassette between F3141-F3140 homology sequences are shown in Supplementary Data 9. *zeoR* marker was eliminated by using an in-lab p15Flip plasmid encoding the Flippase. SacB counterselection was used to eliminate recombineering plasmids. F3141-F3140 (*apsAB*) operon deletion were confirmed by PCR. The absence of mutation in the ST38_1Δ*apsAB* clone used for experimental evolution was determined by WGS. For complementation, recipient cells were electroporated with the plasmids pHV7-empty, pHV7-F3141, pHV7-F3140, pHV7-F3141-F3140 and pHV7-F3141-F3140-mutants (Supplementary Data 9). pHV7-derivatives carrying strains were incubated 24 h in LB, apramycin 50 μg ml-1, 0.4% glucose to repress $p_{BAD}$ activity ($t_0$); two serial passages using 1:200 dilutions were then performed in LB, apramycin 50 μg ml-1 and 0.02% arabinose to induce F3141-F3140 transcription. Cultures were diluted and plated on LB agar with or without MEM 0.05 μg ml-1 and pOXA-48 plasmids stability assessed as the ratio of resistant colonies over whole population.

## Carbapenem susceptibility testing

MEM Minimal Inhibitory concentration (MIC) was determined by Etest (Biomerieux). The plates were inoculated by flooding 2 ml of bacterial culture ($10^6$ bacteria ml-1), spread by a gentle rocking motion, and

excessive liquid was removed leaving 0.5 ml of culture (+/− 10%). The flooding method was selected over swab streaking as it provides a more accurate reading of the Etest result.

## Bacteriophages plaque assays
Phage plaque assays have been performed as previously described and by using the same phage collection[28] (bacteriophages lambda, T4, P1, 186cIts, CLB_P2, LF82_P8, AL505_P2, and T5, Supplementary Table 1). Phages were obtained as active cultures. The preys, *E. coli* K12 MG1655 strain and its isogenic derivatives chromosomally encoding the miniTn*apsAB* or the miniTn*zeoR* as control were grown overnight. Overnight cultures were diluted to 1:20 in the presence of arabinose 0.2% in LB medium and incubated at 37 °C for 3 h to induce *apsAB* expression. Bacterial lawns were prepared by mixing 200 µL of the induced culture with 100 µL of CaCl$_2$ 1 M and 20 ml of LB + 0.5% agar and poured onto 12 × 12 cm square plates of LB containing 0.2% arabinose. High-titer (>10$^8$ pfu ml-1) stocks of phages lambda, T4, P1, 186cIts, CLB_P2, LF82_P8, AL505_P2, and T5 serially diluted were spotted on each plate and incubated at 37 °C overnight except for phage T7 incubated overnight at room temperature.

## Whole genome sequencing and mutation identification
ST38_1, ST38_2 and ST38_3 strains were fully sequenced and used as reference sequences by combining Illumina sequencing and PacBio sequencing (ST38_1, ST38_2) or Oxford Nanopore technology (ST38_3). DNA was extracted at the exponential phase with Qiagen Puregene Yeast/Bact kit B. PacBio sequencing libraries were prepared with NANOBIND CBB KIT RT PacBio. ST38_3 Nanopore sequencing library was prepared by using Native Barcoding Kit 24 V14 (ref SQK-NBD114.24), and sequencing was performed with flowcell R10.4.1 (ref FLO-MIN114) on MinION Mk1C device with Guppy 5.0.14 as a base-caller. Long-read PacBio sequences were assembled with hybrid-SPAdes v 3.15.5 for hybrid assembly of short and long reads[42]. Hybrid assembly of short and long-read Nanopore sequences were performed by using Canu 2.2[43] and Circlator 1.5.5[44].

For Illumina sequencing, DNA was extracted from stationary phase cultures by using the Qiagen Blood and Tissue DNeasy kit, libraries were prepared with the NEBNext Ultra II FS DNA Library Prep Kit and sequencing was performed with NovaSeq6000 or NextSeq500 sequencing platforms. Short-read Illumina sequences were assembled with SPAdes 3.15.5[45] or aligned to the reference sequences by using Breseq 0.35.7[46] to identify SNPs, deletions, insertions and recombination events. IGV 2.11.9[47] was used to visually confirm mutation events. Illumina-reads of the whole population of the evolved lineages collected at day 7, day 14, day 21, and day 28 were analyzed for new junction evidence and coverage distribution by using Breseq 0.35.7[46] with the -p option for pool sequencing, a polymorphism frequency cutoff 2.5% option with at least 10 polymorphic reads. Plasmid coverage inside and outside the Tn*6237* sequence was determined as the number of reads on three 15 kb-long regions, one in Tn*6237*, two outside, containing no IS, by using BAM files and the GRanges function of the GenomicAlignments package 1.22.1 in RStudio (R 3.6.3). The ratio was calculated as the ratio of the average of the number of reads mapping on the two regions outside Tn*6237* to the number of reads mapping on Tn*6237*. IS*1* new junctions detected by Breseq were tested as potential Tn*6237* integration sites by PCR using a primer located near the new IS*1* insertion site and a primer in *bla*$_{OXA-48}$ (Supplementary Data 9). PCR products were Sanger-sequenced to confirm the insertion site.

## Phylogenetic analysis of *E. coli* ST38 and sequence annotation
To contextualize the three ST38 strains used in this work we performed a phylogenetic analysis using 1907 sequences retrieved from public databases (April 2020): 1248 assembled genome sequences from Enterobase [https://enterobase.warwick.ac.uk/], 149 assembled

genomes from the NCBI and 510 sequences retrieved as reads and assembled with SPAdes 3.12.0[45] (Supplementary Data 10). QUAST 2.2[48] was used to assess the assembly quality and contigs shorter than 500 bp were filtered out for the phylogenetic analysis. A core genome alignment was generated with Parsnp 1.5.4[49], by using a finished genome sequence as reference. Maximum-Likelihood (ML) trees were generated with RAxML 8.2.12[50] using GTRGAMMA after removing regions of recombination with Gubbins[51]. The ST38 single locus variant ST963 strain CNRC6O47 was used as outgroup to root the phylogenetic tree. Trees were visualized and annotated using ITOL[52] [https://itol.embl.de/].

Genomes annotation was performed by using Prokka 1.14.5[53]. Resistome and plasmidome were characterized by using ABRicate 1.0.1 on the ResFinder db[54] (minimum coverage 60%, minimum identity 95%) and PlasmidFinder 2.1.1 (minimum coverage 60%, minimum identity 95%)[55]. Identification of defense systems was performed by using DefenseFinder [https://defensefinder.mdmlab.fr/].

## In silico characterization of ApsAB antiplasmid systems
Search for *apsAB* homologs were performed by using PSI-BLAST with three iterations on the recently introduced NCBI clustered nr database [https://blast.ncbi.nlm.nih.gov/]. This database is composed of representative sequences of clusters, that groups NCBI nr sequences sharing 90% identity and 90% length to other members of the cluster. Only sequences longer than 1100 a.a. residues (ApsA) and 500 a.a. residues (ApsB) were kept for PSI-BLAST iterations. CDS located downstream of *apsA* homologs were retrieved by using GCsnap 1.0.17[56]. Protein sequences were aligned by using MuscleW 3.8.31 (default options) under Jalview 2.11.3[57] and a distance tree was created by Neighbor-Joining method using a BLOSUM62 matrix. Remote homology detection was also performed with the HHpred server [https://toolkit.tuebingen.mpg.de/tools/hhpred] using PDB_mmCIF70_18_jun, SCOPe70_2_08, CATH_S40_v4.3 and UniProt-SwissProt-viral70_3_nov_2021 as target databases accessed in September 2023. Structural modelling was performed with Alphafold2[58] implemented in Neurosnap [https://neurosnap.ai/] and the resulting models were used as templates for similarity searches with Foldseek [https://search.foldseek.com/search]. Structure annotation was performed with Pymol 2.5.5 (The PyMOL Molecular Graphics System, Version 2.5.5 Schrödinger, LLC.). Alpha-fold or icn3d pdb models of 20 ApsB-like or DdmE-like proteins (Supplementary Data 8) were recovered from Uniprot [https://www.uniprot.org/] or ncbi [https://www.ncbi.nlm.nih.gov/Structure/icn3d/] and aligned to ApsB structure model under Pymol 2.5.5 and the Root-mean-square deviation of atomic positions (RMSD) was used as a proxy to estimate protein structure similarity. The genomic environment of *apsAB* homologs from 12 *E. coli* strains belonging to different STs was compared by using the web version of Clinker [https://cagecat.bioinformatics.nl/tools/clinker].

## Statistical tests
All graphs were generated with R (version 3.6.3) by using the R packages (tidyverse, forcats, ggplot2, ggpubr, rstatix, broom). Normality of data were assessed by using the Shapiro test. All the statistical analyzes were performed with a pairwise two sample t.test and p-value were corrected following Benjamini-Hochberg correction (FDR).

## Reporting summary
Further information on research design is available in the Nature Portfolio Reporting Summary linked to this article.

# Data availability
The data that support the findings of this study are provided within the manuscript and the associated supplementary materials. Full details and links to the publicly available databases [https://enterobase.

warwick.ac.uk/], [https://defensefinder.mdmlab.fr/], [https://www.uniprot.org/], [https://www.ncbi.nlm.nih.gov/Structure/icn3d/], [https://www.ncbi.nlm.nih.gov/] used in bioinformatic analyzes are provided in the methods and their associated references. Complete genome sequences of ST38_1, ST38_2, and ST38_3 and of the three pOXA-48 plasmids have been deposited at DDBJ/EMBL/GenBank (BioProject PRJEB71895). The corresponding Genome_ID are provided in Supplementary Data 1 (for ST38 strains) and 9 (for pOXA-48s). Illumina reads for the three *K. pneumoniae* used as donors of pOXA-48 have also been deposited at DDBJ/EMBL/GenBank (BioProject PRJEB71895), and their accession numbers are given in Supplementary Data 1. Sequence data from individual colonies and pools are available from the corresponding author upon request. Source data are provided in this paper.

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

## Acknowledgements

We thank Alexandre Almeida for his critical reading of the manuscript, David Vallenet for his help in the structure analysis of ApsAB, and Jean-Marc Ghigo for providing *E. coli* strain MFDpir. Sequencing was performed at the Biomics Platform, C2RT, Institut Pasteur, Paris, France, supported by France Génomique (ANR-10-INBS-09) and IBISA. This work was supported by "Investissement d'Avenir" program, LABEX IBEID (Grant ANR-10-LABX-62-IBEID) and SEQ2DIAG (ANR-20-PAMR-0010). PDZ received a fellowship from CIOSPB (Center national de l'Information, de l'Orientation Scolaire et Professionnelle, et des Bourses) of the Ministry of Higher Education, Scientific Research and Innovation of Burkina Faso and from the LABEX IBEID.

## Author contributions

I.R.C., P.G. and P.D.Z. designed the study. P.D.Z., I.R.C., NC, F.D. and G.R. performed the experiments. P.D.Z., P.G. and I.R.C. analyzed the data. F.D., T.N. and A.H. provided materials. I.R.C., P.G. and P.D.Z. wrote the manuscript with input from G.R. All authors read and approved the final manuscript.

## Competing interests

All authors declare no competing interests.
