## [Peer Review File · Nature Communications]

REVIEWERS' COMMENTS

Reviewer #1 (Remarks to the Author):

“A novel antiplasmid system drives antibiotic resistance gene integration in 2 carbapenemase-producing *Escherichia coli* lineages”

The manuscript is a very interesting study on how resistance genes present on plasmids can end up integrated in the chromosome to spare bacteria the cost of carrying the plasmid.

The study is biologically sound and very thorough and tackles a very interesting biological problem. The results are noteworthy and fill a gap in our knowledge in the field.

The fact that OXA-48 resistance gene got integrated into the *E. coli* ST38 chromosome is a well known fact and there are plenty of papers describing it epidemiologically (<https://www.ncbi.nlm.nih.gov/pmc/articles/PMC3186974/> . <https://www.microbiologyresearch.org/content/journal/jmm/10.1099/jmm.0.000248>) but I had never come across a paper attempting to understand why this happened, using experimental evolution. Which is what this paper does.

The work they do in experimental evolution is extensive, using 3 ST_38 alleles and 3 p-OXA48 variants. This allows the authors to do comparative genomics on the different results of the different transconjugants they obtain and finally pinpoint the genes that make this integration possible.

The methodology is sound, and I do not think that it needs more experiments or analysis.

More details must be integrated into the manuscript to make it enjoyable. I needed to go and dig some details from Supplementary table, while they should be clearly stated on the main text. In particular:

- Can the authors give more detail on the 3 *E. coli* ST38 samples they choose to use in their experiments? Please state clearly in the main text that while there are many ST38 that carry OXA-48 integrated in their genomes, the ones you choose do not. Can you comment more on O-types/H-types, what does it mean and why they did influence your choice of strains? Please comment why you choose these specific 3 samples, if they have any difference that you might find interesting, or maybe a different origin. Also, please comment on their position on the ST38 phylogenetic tree (I think that this choice is very good on your part, but it is very hard for a busy reader to grasp the relevance of you having 3 samples of ST38 in your analysis and not just one). In methods you gave some characterization of the plasmids present in ST38_1 and ST38_3, but not of ST38_2. Does it mean ST38_2 did not have any plasmid (or any relevant plasmids)? If so, please state clearly, and describe why you chose it.

- Can the authors give more details on the 3 p-OXA-48 variants used in the experiments? In particular p-OXA1 and p-OXA3 in figure 1a seem identical, except maybe for the transposon inserted upside down. Could you explain all these details more clearly in the main text as they show that you really thought

your experiment through with great care? Can you please comment how the presence of one or two IS1 and the structure of Tn6237 could affect your experiments?

- The authors find that ApsAB is a key factor for OXA-48 gene insertion in the chromosome. Can the authors comment on whether these genes are present in the ST38 strains that have been already described in literature as carrying OXA48 on their chromosome?

- Can you please leave “transconjugants” and “duplication time” written in full instead of “TC” and “DT”?

Reviewer #2 (Remarks to the Author):

The authors present a detailed and comprehensive study that ultimately identified a novel antiplasmid system, ApsAB, via the analysis of convergent evolutionary mutations. The conclusions of the study are well-supported by the experimental data. The authors undertook a detailed series of experiments, which incorporated traditional fitness and plasmid stability testing, whole-genome sequencing, mutational analysis and complementation, and a bioinformatics-based investigation of *apsAB* homologs.

The findings of this study are highly significant. The findings are applicable in not just understanding plasmid maintenance and stability, but also in the potential for this system to be used in a guide-based manner to cure plasmids. The system appears to have a relatively broad host range, targeting both high and low-copy number plasmids, and future work will likely determine the complete host range and the underlying guided targeting mechanism. As alluded to in the Discussion, this system may also represent a broader diverse family of defence systems acting on plasmids, so the implications could be wide-ranging.

The methods are detailed and include all required controls.

The manuscript is well-written, and I only have minor comments to improve the manuscript.

- 1) Line 79-80. The sentence is dis-jointed and should to be re-written.
- 2) Line 168-169. As above, dis-jointed sentence.

Reviewer #3 (Remarks to the Author):

In the manuscript ‘A novel antiplasmid system drives antibiotic resistance gene integration in carbapenemase-producing *Escherichia coli* lineages’ the authors describe how they use experimental evolution of three different *E. coli* ST38 strains combined with three different *bla*OXA-48 -carrying plasmids to investigate the mobilization of *bla*OXA-48 from the plasmid to the chromosome. The authors observe that chromosomal integration of *bla*OXA-48 is dependent on plasmid-induced fitness cost, a mobile genetic structure embedding the antimicrobial resistance gene, and a novel antiplasmid system (ApsAB).

It is an impressive study that describes an interesting and relevant finding in the field, however the manuscript lacks a clear aim and conclusion and the results can be difficult to follow due to the comprehensive amount of results generated. This section needs attention and revision from the authors to make it more readable for the recipient. The introduction as well as materials and methods are well constructed and thorough.

Included here are major comments and specific minor comments that can improve the manuscript, which I will be happy to review again, in a refined version.

Major comments

When reading the result section, it is not clear exactly which strain/plasmid combinations have been included in which experiments and why only certain results are shown in the figures and discussed further. An example of this is Fig. 2e where results are shown for ST38_2 pOXA-48_1 MEM / 1 day and MEM / 3 days while for ST38_3 pOXA-48_1 only results for MEM / 1 day are shown. A flow diagram explaining exactly which experiments were done on which strain/plasmid combinations could be added to make this clearer and give an overview of the entire study.

A table describing the strain/plasmid combinations with unique names, as listed in supplementary Dataset for Fig. 1b, could help track them in the results.

Furthermore, the result section takes up an unproportionable large part of the paper compared to the discussion and there are parts in the result section that belong in the discussion. See for example lines 142-142, 155-157 and 185-186.

In multiple figures you have a y-axis showing the Percentage of plasmid carrying colonies (%) and in many of these the percentage is above 100. It is not clear how more than all of the colonies can carry the plasmid.

Fig. 5 shows how no antiphage activity was observed when ApsAB was chromosomally expressed in MG1655 against eight different phages. It is not particularly relevant to the rest of the study and should be moved to the supplementary material.

In general, the amount of figures is too much. Could some of the results be presented in different ways, perhaps tables? Or perhaps sort the figures and only keep the most relevant ones.

The authors mention different types of defense systems in the discussion and in relation to this it would be interesting to see a characterization of which systems are present in the ST38 strains used in this study and a discussion of whether this has an influence on the results.

Specific minor comments

There is no figure legend for Fig. 3f.

Supplementary Fig. 2 is of very poor quality.

Line 77: Please add that it is IncL plasmids.

Line 117-119: The authors write that there are five integrations in the IncFII plasmid but in supplementary Fig. 3, seven integration sites are marked.

Line 165: It is unclear which plasmid you refer to when you write pOXA-48. This is general throughout the manuscript.

Line 187: Please specify which mutations in the three ST38_2.

Line 216: Why did you choose to test the ColE1 plasmid of the five different plasmids?

Line 229: Z103 is the name of the ST38_1/pOXA-48_1 TC making 'ST38_1' and 'pOXA-48_1' in ST38_1Z103ΔapsAS/pOXA-48_1 unnecessary. As stated in major comment, it could be easier to follow if you used these names throughout the manuscript.

Line 311-312: What do you mean by 'reunify' DdmDE- and ApsAB-like systems? Please provide a reference and correct background to describe how they were divided.

Line 476: Please specify base call model.

Line 512: Please provide information on minimum coverage and identity for PlasmidFinder results.

Line 807: Is it ca. 10 copies or ca. 15-20 copies?

Reviewer #4 (Remarks to the Author):

POINT-BY-POINT RESPONSE TO REVIEWERS' COMMENTS

Reviewer #1 (Remarks to the Author):

“A novel antiplasmid system drives antibiotic resistance gene integration in 2 carbapenemase-producing *Escherichia coli* lineages”.

The manuscript is a very interesting study on how resistance genes present on plasmids can end up integrated in the chromosome to spare bacteria the cost of carrying the plasmid.

The study is biologically sound and very thorough and tackles a very interesting biological problem. The results are noteworthy and fill a gap in our knowledge in the field. The fact that OXA-48 resistance gene got integrated into the *E. coli* ST38 chromosome is a well known fact and there are plenty of papers describing it epidemiologically (<https://www.ncbi.nlm.nih.gov/pmc/articles/PMC3186974/> . <https://www.microbiologyresearch.org/content/journal/jmm/10.1099/jmm.0.000248>) but I had never come across a paper attempting to understand why this happened, using experimental evolution. Which is what this paper does. The work they do in experimental evolution is extensive, using 3 ST_38 alleles and 3 p-OXA48 variants. This allows the authors to do comparative genomics on the different results of the different transconjugants they obtain and finally pinpoint the genes that make this integration possible. The methodology is sound, and I do not think that it needs more experiments or analysis.

We thank the reviewer for accepting the task to assess our work and for his/her helpful comments.

More details must be integrated into the manuscript to make it enjoyable. I needed to go and dig some details from Supplementary table, while they should be clearly stated on the main text.

In particular:

1.1 Can the authors give more detail on the 3 *E. coli* ST38 samples they choose to use in their experiments? Please state clearly in the main text that while there are many ST38 that carry OXA-48 integrated in their genomes, the ones you choose do not. Can you comment more on O-types/H-types, what does it mean and why they did influence your choice of strains? Please comment why you choose these specific 3 samples, if they have any difference that you might find interesting, or maybe a different origin. Also, please comment on their position on the ST38 phylogenetic tree (I think that this choice is very good on your part, but it is very hard for a busy reader to grasp the relevance of you having 3 samples of ST38 in your analysis and not just one). In methods you gave some characterization of the plasmids present in ST38_1 and ST38_3, but not of ST38_2. Does it mean ST38_2 did not have any plasmid (or any relevant plasmids)? If so, please state clearly, and describe why you chose it.

The reviewer points a key point of our work we did not explained enough. To set up a model for blaOXA-48 integration and reveal a possible role of the genetic background, we chose three ST38 isolates from water or sewage that did not carry bla_{OXA-48}. They differed by their position in the ST38 phylogenetic tree (Fig. S1). In particular, ST38-1 belonged to a branch of the tree also encompassing three sub-clades characterized by different bla_{OXA-48} chromosomal integrations (Fig. S1). It shared with one of these sub-clades the same H-and O-antigens (O2-H30). Plasmid were scarce in the three isolates. ST38_1 contained only a IncFII plasmid, ST38_3 a IncFIC(FII) plasmid and ST38_2 had no plasmid. These three isolates were multiresistant. Their complete genome sequence showed that in ST38_1 and _2, all ARGs were chromosomally integrated. In contrast in ST38_3 most ARGs were carried by the IncFIC(FII) plasmid.

We have now introduced this explanation in the manuscript (Results section, Lines 72 to 85).

1.2 Can the authors give more details on the 3 p-OXA-48 variants used in the experiments? In particular p-OXA1 and p-OXA3 in figure 1a seem identical, except maybe for the transposon inserted upside

down. Could you explain all these details more clearly in the main text as they show that you really thought your experiment through with great care? Can you please comment how the presence of one or two IS1 and the structure of Tn6237 could affect your experiments?.

We explained the differences between the three plasmids in the legend of Fig. 1. However, the reviewer is right that it is important to further detail the presence of IS1 and of Tn6237 in the main text in addition to the legend of Fig. 1. These explanations have been introduced Lines 88 to 93.

1.3 The authors find that ApsAB is a key factor for OXA-48 gene insertion in the chromosome. Can the authors comment on whether these genes are present in the ST38 strains that have been already described in literature as carrying OXA48 on their chromosome?

ApsAB is present in the three strains used in the study as well as numerous ST38 strains, as annotated in the outside circles of Fig. S1. It is present in the four ST38 lineages in which *bla*_{OXA-48} is integrated in the chromosome (colored in blue, red, braun and green in Supplementary Fig. 1). We have now discussed this point in the Discussion section, Lines 349 to 352.

1.4 Can you please leave “transconjugants” and “duplication time” written in full instead of “TC” and “DT”?

We have replaced TC and DT by “transconjugants” and “doubling time” throughout the manuscript as suggested.

Reviewer #2 (Remarks to the Author):

The authors present a detailed and comprehensive study that ultimately identified a novel antiplasmid system, ApsAB, via the analysis of convergent evolutionary mutations. The conclusions of the study are well-supported by the experimental data. The authors undertook a detailed series of experiments, which incorporated traditional fitness and plasmid stability testing, whole-genome sequencing, mutational analysis and complementation, and a bioinformatics-based investigation of *apsAB* homologs.

The findings of this study are highly significant. The findings are applicable in not just understanding plasmid maintenance and stability, but also in the potential for this system to be used in a guide-based manner to cure plasmids. The system appears to have a relatively broad host range, targeting both high and low-copy number plasmids, and future work will likely determine the complete host range and the underlying guided targeting mechanism. As alluded to in the Discussion, this system may also represent a broader diverse family of defence systems acting on plasmids, so the implications could be wide-ranging.

The methods are detailed and include all required controls.

The manuscript is well-written, and I only have minor comments to improve the manuscript.

We thank the reviewer for assessing our work and for his/her positive comments.

2.1 Line 79-80. The sentence is dis-jointed and should to be re-written.

Following the request from Reviewer 1, we have further described the three plasmids and completely rewritten this part (Lines 88 to 93).

2.2 Line 168-169. As above, dis-jointed sentence.

We have split this sentence in two sentences and it reads as follows (Lines 195 to 197): Large chromosomal deletions ranging from 15 to 40 kb and encompassing *mutS* and *rpoS* were also observed in evolved ST38_1 lineages. They were independent on pOXA-48 presence as they also occurred in control experiments.

Reviewer #3 (Remarks to the Author):

In the manuscript 'A novel antiplasmid system drives antibiotic resistance gene integration in carbapenemase-producing *Escherichia coli* lineages' the authors describe how they use experimental evolution of three different *E. coli* ST38 strains combined with three different blaOXA-48 -carrying plasmids to investigate the mobilization of blaOXA-48 from the plasmid to the chromosome. The authors observe that chromosomal integration of blaOXA-48 is dependent on plasmid-induced fitness cost, a mobile genetic structure embedding the antimicrobial resistance gene, and a novel antiplasmid system (ApsAB).

It is an impressive study that describes an interesting and relevant finding in the field, however the manuscript lacks a clear aim and conclusion and the results can be difficult to follow due to the comprehensive amount of results generated. This section needs attention and revision from the authors to make it more readable for the recipient. The introduction as well as materials and methods are well constructed and thorough.

Included here are major comments and specific minor comments that can improve the manuscript, which I will be happy to review again, in a refined version.

We thank the reviewer for his/her comments and suggestions to make the results and discussion section easier to follow and to bring a clear aim to our work. As stated by the reviewer, we provided comprehensive data to characterize the factors contributing to the integration of blaOXA-48 and to demonstrate the major role of the antiplasmid system we have discovered. To reach this goal, we have performed numerous and complementary evolution experiments with different strains and different plasmids. We apologized if it seemed to be somewhat confusing. We have now tried to better explain the objective of each evolution experiment et to recapitulate the experiments in a Table (Table 1) to be part of the main text. We believe that by responding to his or her major and specific comments and also to comments from reviewers 1 we have made our demonstration convincing and we have better highlighted the novelty of our work.

Major comments

3.1 When reading the result section, it is not clear exactly which strain/plasmid combinations have been included in which experiments and why only certain results are shown in the figures and discussed further. An example of this is Fig. 2e where results are shown for ST38_2 pOXA-48_1 MEM / 1 day and MEM / 3 days while for ST38_3 pOXA-48_1 only results for MEM / 1 day are shown. A flow diagram explaining exactly which experiments were done on which strain/plasmid combinations could be added to make this clearer and give an overview of the entire study. A table describing the strain/plasmid combinations with unique names, as listed in supplementary Dataset for Fig. 1b, could help track them in the results.

Indeed, we have performed five evolution experiments, each, with a specific objective to characterize the factors contributing to the integration of *bla*_{OXA-48} in ST38 *E. coli* strains. As suggested by the reviewer, we have added a new table (table 1) which summarizes all the experiments performed and their objective. This table also refers to the figure showing the results. We hope this will be helpful to follow the experimental design.

3.2 Furthermore, the result section takes up an unproportionable large part of the paper compared to the discussion and there are parts in the result section that belong in the discussion. See for example lines 142-142, 155-157 and 185-186.

We agree with the reviewer that the results section is detailed, then leaving less place for a long discussion. We thank the reviewer for pointing a few sentences of the results section that are more appropriate for the discussion. We have moved these points to the discussion (Line 320 to 326 and 332 to 336).

3.3 In multiple figures you have a y-axis showing the Percentage of plasmid carrying colonies (%) and in many of these the percentage is above 100. It is not clear how more than all of the colonies can carry the plasmid.

We thank the Reviewer for pointing this out. In fact, we plated 100 µl of diluted cultures on selective and non-selective plates followed by automatic numeration with the SCAN4000. Due to plating biases, there might be few more colonies on the plate with meropenem comparing to the plate without. As we performed a ratio of the number of colonies on the selective plate divided by the number of colonies on the non-selective plate, the ratio might be above 100%. We have now explained percentages higher than 100% in the Methods section (Lines 460-461).

3.4 Fig. 5 shows how no antiphage activity was observed when ApsAB was chromosomally expressed in MG1655 against eight different phages. It is not particularly relevant to the rest of the study and should be moved to the supplementary material.

We agree with the Reviewer and we have moved this figure as a supplementary figure (Supplementary Fig. 4)

3.5 In general, the amount of figures is too much. Could some of the results be presented in different ways, perhaps tables? Or perhaps sort the figures and only keep the most relevant ones.

Given the number of experiments, we feel necessary to keep a graphical representation of the results through figures. We have reorganized the figure to improve their readability. We have removed Figure 5 describing the phage susceptibility experiments and fused the Fig. 8 and 9 related to the ApsAB system. On the other hand, we have split the former Fig. 2 (now Fig. 2 and 3) to separate the different stages of our analysis of the factors contributing to the integration of *bla*_{OXA-48}. In addition, we believe that the new Table 1 with reference to the figures improves the readability.

The authors mention different types of defense systems in the discussion and in relation to this is would be interesting to see a characterization of which systems are present in the ST38 strains used in this study and a discussion of whether this has an influence on the results.

A description of other defense systems identified in the three strains has been added in supplementary Table 1. Nine to 11 defense systems, as identified by using Defense Finder, are predicted in the three

strains (9 in ST38_1; 11 in ST38_2 and 10 in ST38_3). Most of them are shared by two or the three strains. A restriction-modification system unique to ST38_2 might have affected our capacity to genetically modify ST38_2, as we failed in this strain to use recombineering methods that worked in ST38_1 and ST38_3. Mutations in these other defence systems were not selected in the course of the experimental evolution. Therefore, we have no evidence that they could have impacted the results of the evolution experiments. We have now discussed the presence of other defense systems in the three strains Lines 346 to 348. This stands as follows: “In addition to ApsAB, nine to eleven other defense systems were predicted in the three strains (Supplementary Data 1). However none was specific to ST38_1 and we did not detect any mutation in these systems during experimental evolutions. They probably did not influence the evolution results.”

Specific minor comments

3.6 There is no figure legend for Fig. 3f.

The figure legend for Fig. 3f is now added.

3.7 Supplementary Fig. 2 is of very poor quality.

As this figure presents captures from Breseq results, it was not possible for us to increase the quality of the definition. However, we believe that despite its quality, this supplementary figure provides a visual estimate of the sequencing coverage along the plasmid sequence quantified in figure 2b and figure 6d.

3.8 Line 77: Please add that it is IncL plasmids.

We added the incompatibility group of the plasmid (Line 75).

3.9 Line 117-119: The authors write that there are five integrations in the IncFII plasmid but in supplementary Fig. 3, seven integration sites are marked.

We thank the Reviewer to have pointed this error. In a previous version of our manuscript, we provided the number of integrations determined from the two experimental evolutions with ST38_1/pOXA-48_1 (experimental evolutions 1 and 2). In the final version of the manuscript, only the results of the experimental evolution 1 were detailed in the text. We forgot to accordingly correct the figure and the number of integrations cited in the text. Following experimental evolution 1, there were three integrations in the plasmid. The figure now displaced to Fig. 2c) and the text (Lines 139) have been corrected.

3.10 Line 165: It is unclear which plasmid you refer to when you write pOXA-48. This is general throughout the manuscript

We found eight mutations in pOXA-48_1 and one in pOXA-48_2. This has now been introduced in the corresponding sentence (Line 193).

We generally used pOXA-48 to designate the three pOXA-48 plasmids, not referring to one plasmid in particular. In the revised version we have mentioned the variant, or replace pOXA-48 by pOXA-48 plasmids, when appropriate.

3.11 Line 187: Please specify which mutations in the three ST38_2.

The sentence stands now as: Three ST38_1 and three ST38_2 evolved transconjugants mutated in F3140 or in F3141 were tested...(Lines 214, 215)

3.12 Line 216: Why did you choose to test the ColE1 plasmid of the five different plasmids?

We chose ColE1 plasmid because it shows a higher copy number and it was stable after five passages without selective pressure when ApsAB is not expressed (Fig 5a). Therefore, the rapid loss of ColE1 plasmid due to expression of ApsAB after a 3 to 6h induction shows the efficiency of the system to eliminate this plasmid.

3.13 Line 229: Z103 is the name of the ST38_1/pOXA-48_1 TC making 'ST38_1' and 'pOXA-48_1' in ST38_1Z103ΔapsAS/pOXA-48_1 unnecessary. As stated in major comment, it could be easier to follow if you used these names throughout the manuscript.

We agree that ST38_1Z103ΔapsAS is an unnecessary precision and we have replaced it by ST38_1ΔapsAS (in particular Line 254, 256, in Table 1 and in the legend of Fig. 6). However, we believe that the naming strain/plasmid is the most appropriate in the context of the different combinations we have used.

3.14 Line 311-312: What do you mean by 'reunify' DdmDE- and ApsAB-like systems? Please provide a reference and correct background to describe how they were divided.

We agree with the Reviewer that the term "reunify" was not appropriate. Therefore, we replaced it by the word "connect" (Line 363).

3.15 Line 476: Please specify base call model.

The basecaller that was used was Guppy 5.0.14. We have now added this information in the Methods section (Line 530)

3.16 Line 512: Please provide information on minimum coverage and identity for PlasmidFinder results.

We added the information regarding minimum coverage and identity for PlasmidFinder Lines 567-568 (minimum coverage 60%, minimum identity 95%).

3.17 Line 807: Is it ca. 10 copies or ca. 15-20 copies?

We thank the Reviewer for pointing this error. Some words were missing. We have now carefully revised the legend and corrected the mistakes (Lines 894 to 896).

Reviewer #4 (Remarks to the Author):
